# TRADING-OFF MULTIPLE PROPERTIES FOR MOLECULAR OPTIMIZATION

## ABSTRACT

Molecular optimization, a critical research area in drug discovery, aims to enhance the properties or performance of molecules through systematic modifications of their chemical structures. Recently, existing Multi-Objective Molecular Optimization (MOMO) methods are extended from Single-Objective Molecular Optimization (SOMO) approaches by employing techniques such as Linear Scalarization, Evolutionary Algorithms, and Multi-Objective Bayesian Optimization. In Multi-Objective Optimization, the ideal goal is to find Pareto optimal solutions over different preferences, which indicate the importance of different objectives. However, these straightforward extensions often struggle with trading off multiple properties due to the conflicting or correlated nature of certain properties. More specifically, current MOMO methods derived from SOMO are still challenged in finding preference-conditioned Pareto solutions and exhibit low efficiency in Pareto search. To address the aforementioned problems, we propose the **P**reference-**C**onditioned **I**nversion (PCI) framework, efficiently "inverting" a pre-trained surrogate oracle under the guidance of a non-dominated gradient, to generate candidate Pareto optimal molecules over preference-conditioned distributions. Additionally, we provide theoretical guarantees for PCI's capability in converging to preference-conditioned solutions. This unique characteristic enables PCI to search the full Pareto front approximately, thereby assisting in the discovery of diverse molecules with varying ratios of properties. Comprehensive experimental evaluations show that our model significantly outperforms state-of-the-art baselines in multi-objective molecular optimization settings.

## 1 INTRODUCTION

Molecular optimization in drug design usually involves reasoning about multiple, often conflicting or correlated, objectives (Fromer & Coley, 2023). For example, for a new drug to be successful, it must simultaneously be potent, bioavailable, safe, and synthesizable (Dara et al., 2022). More specifically, a realistic drug customization scenario requires maximizing the combined stability and solubility (Gupta et al., 2004), in which two properties may inherently conflict. Generally, these objectives exhibit implicit relationships, which can be either conflicting or correlated, rather than being independent. Consequently, it is infeasible to find a single molecule that maximizes all objectives simultaneously. Therefore, in Multi-Objective Optimization, the ideal goal is to identify Pareto solution set that cover all the possible trade-offs among objectives (Miettinen, 1999; Ehrgott, 2005).

Recently, significant progress has been made in **Single-Objective Molecular Optimization** (SOMO). This naturally prompts the question: Can classical SOMO methods be readily extended to **Multi-Objective Molecular Optimization** (MOMO) scenarios? Most of the existing works, including Constrained Generative Models (CGM) (Gómez-Bombarelli et al., 2018; Jin et al., 2018; Griffiths & Hernández-Lobato, 2020; Wang et al., 2023; De Cao & Kipf, 2018; Shi et al., 2020; Liu et al., 2021; Shi et al., 2020) and Combinatorial Optimization (CO) algorithms (Fu et al., 2022; You et al., 2018; Ståhl et al., 2019; Zhou et al., 2019; Jin et al., 2020; Gottipati et al., 2020; Jain et al., 2023; Jensen, 2019; Nigam et al., 2020; Chen et al., 2021; Xie et al., 2021; Fu et al., 2021; Korovina et al., 2020), employ the *Linear Scalarization* technique to adapt their methods to MOMO. It assigns a set of weights (preferences) $w_i$ to the objectives $\mathcal{L}_i$, thereby reducing the problem to a single unified objective: $\mathcal{L} = \sum_{i=1}^{m} w_i \mathcal{L}_i$. In addition, *Multi-Objective Evolutionary Algorithms* have been introduced to tackle MOMO (Abbasi et al., 2022). Most of classic multi-objective evolutionary

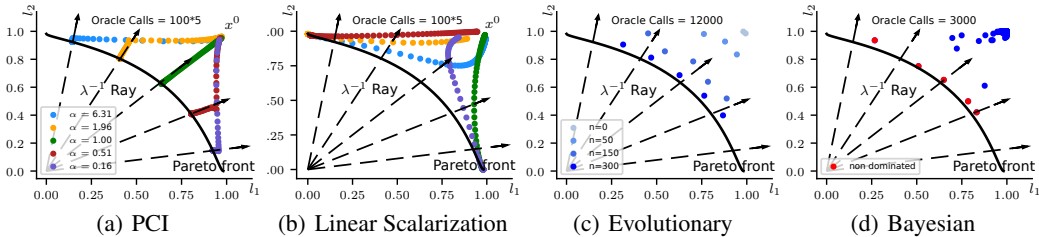

Figure 1: Pareto front (black solid curve) for two loss functions $l_1, l_2$ and solutions (circles) and Oracle calls (computational cost) for different preferences $\alpha = \frac{\lambda_1}{\lambda_2}$ (dashed rays) obtained by (a) PCI: can find solutions at any given preference vector; (b) Linear Scalarization: often results in baised solutions; (c) Multi-Objective Evolutionary Algorithms: using reference vectors to find solutions but fail to capture the preference condition and with high computational cost; (d) Multi-Objective Bayesian Optimization: using expected hypervolume improvement as acquisition function, with no preference guidance and high computational cost. More details can be found in Section. 5.1.

algorithms (Deb et al., 2002; Deb & Jain, 2013; Coello & Lechuga, 2002) employ strategies such as crowding distance sorting or reference points to make solutions spread out along the Pareto front. Furthermore, several methods based on *Multi-Objective Bayesian Optimization* (Shah & Ghahramani, 2016) have been introduced to MOMO (Jain et al., 2023; Fromer & Coley, 2023), leveraging the expected or probability of hypervolume improvement as an acquisition function to generate a diverse range of Pareto optimal solutions.

Despite existing SOMO methods can be extended to MOMO scenarios, they exhibit several limitations in identifying the Pareto optimal solution set: **(1) Fail to capture preference-conditioned Pareto optimal solutions, which is the intersection point between the Pareto front and preference ray. This limitation may leave certain regions of the Pareto front unexplored, and fail to search the full Pareto front.** The Linear Scalarization often fails to capture specific trade-offs and results in biased solutions, which has been mathematically analyzed by Boyd et al. (2004), see Figure. 1(b). When varying preference vectors, the solutions tend to cluster at the ends of the Pareto front and leave certain regions unexplored. Although Multi-Objective Evolutionary Algorithms and Multi-Objective Bayesian Optimization are guaranteed to obtain Pareto optimal set as the number of iterations approaching infinity, they are often suboptimal within finite optimization steps (Schwefel, 1993; Lam et al., 2016), see Figure. 1(c) and 1(d). Without preference guidance, they also struggle to identify preference-conditioned solutions and fail to cater to chemists' specific drug design needs. **(2) Low efficiency in identifying diverse Pareto optimal solutions.** To capture different trade-offs, CO algorithms with Linear Scalarization often require retraining from scratch for different preferences, thereby amplifying the already high computational cost of CO by a multiple of the number of preferences, which is unacceptable. In addition, Evolutionary Algorithms and Bayesian Optimization also require massive numbers of oracle calls (Bäck et al., 1997; Shahriari et al., 2015).

To address the aforementioned limitations, we propose the Preference-Conditioned Inversion (PCI) framework to identify preference-conditioned Pareto optimal solutions in discrete space, see Figure. 1(a). By adopting the inversion framework, PCI can be precisely guided by the non-dominated gradient w.r.t the chemical structure and adjust the chemical structure towards better uniformity and dominating properties. This unique specific characteristic of the PCI allows for an approximate search of the full Pareto front when the preference vectors cover the space, thereby assisting in the discovery of diverse molecules with varying property ratios. To address the low-efficiency issue, the inversion framework significantly decreases the demand of oracle calls introduced by Pareto search. To better understand our PCI approach, we offer theoretical analysis to demonstrate why PCI is guaranteed to identify preference-conditioned Pareto-optimal solutions in discrete chemical space. The main contributions of our work are:

- We propose the Preference-Conditioned Inversion (PCI) framework to find preference-conditioned Pareto optimal molecules and enable a more efficient full Pareto front search.

- We provide theoretical analysis to demonstrate why PCI can identify preference-conditioned Pareto-optimal solutions in discrete chemical space.

- We evaluate PCI in preference-conditioned and Pareto search MOMO. Comprehensive experimental results demonstrate that PCI significantly outperforms all baseline methods.

## 2 RELATED WORK

**Molecular Optimization.** In recent years, machine learning has shown to be promising for SOMO tasks. Existing methods can be categorized as CGM and CO approaches. CGMs model the molecular distribution with deep generative networks such as VAE (Gómez-Bombarelli et al., 2018; Liu et al., 2018; Jin et al., 2018; 2019; Skalic et al., 2019; Fu et al., 2020; Griffiths & Hernández-Lobato, 2020; Wang et al., 2023), GAN (Guimaraes et al., 2017; De Cao & Kipf, 2018; Abbasi et al., 2022), Flow (Shi et al., 2020), Energy (Liu et al., 2021) and Diffusion-based model (Lee et al., 2023), projecting input molecules into a latent space. CGMs conduct optimization in latent space and reconstruct it to obtain the optimized molecules. However, obtaining the ideal smooth and discriminative latent space, typically required by CGMs, has proven to be a challenge in practice (Brown et al., 2019; Huang et al., 2021). Another research line based on CO directly searches for desired molecules in the explicit discrete space, e.g., Reinforcement Learning (You et al., 2018; Ståhl et al., 2019; Zhou et al., 2019; Jin et al., 2020; Gottipati et al., 2020; Jain et al., 2023), Evolutionary Algorithms (Jensen, 2019; Nigam et al., 2020; Chen et al., 2021), Markov Chain Monte Carlo (Xie et al., 2021; Fu et al., 2021), Tree Search (Fu et al., 2022) and Bayesian Optimization (Korovina et al., 2020; Moss et al., 2020). CO algorithms require massive numbers of oracle calls, which is computationally inefficient during the inference time. Current MOMO methods derived from SOMO, including both CGM and CO approaches, are still challenged in finding preference-conditioned Pareto solutions and exhibit low efficiency in Pareto search.

**Multiple Objective Optimization (MOO) Techniques.** The Pareto optimal solution is highly valuable for MOO since identifying that satisfies all objectives is often impractical. Two primary categories have been extensively studied: Black-Box and White-Box MOO. White-Box MOO methods assume complete knowledge of the inference model and are often based on gradient optimization. MGDA (Désidéri, 2012) is proposed to identify Pareto optimal solutions for low-dimensional data and is extended to high-dimensional scenarios by Sener & Koltun (2018). Subsequently, several efficient methods (Lin et al., 2019; Zhang & Golovin, 2020; Ma et al., 2020) have been proposed to alleviate the negative effects introduced by Linear Scalarization in MGDA. Also, the EPO (Mahapatra & Rajan, 2020) has been developed to find the exact Pareto optimal solution under specified objective preferences. In Black-Box MOO, we assume limited knowledge about the inference model. Therefore, most of the Black-Box MOO approaches are based on Evolutionary Algorithm and Multi-Objective Bayesian Optimization. Multi-Objective Evolutionary Algorithm like NSGA-II (Deb et al., 2002), NSGA-III (Deb & Jain, 2013) and MOPSO (Coello & Lechuga, 2002) employ strategies such as crowding distance sorting or reference points to make solutions spread out along the Pareto front. Also, Multi-Objective Bayesian Optimization has been developed (Shah & Ghahramani, 2016), leveraging the expected or probability of hypervolume improvement as an acquisition function to generate a diverse range of Pareto optimal solutions. However, in gradient-based White-Box MOO, most efforts are concentrated on continuous optimization, ignoring the discrete spaces, especially the complex discrete chemical space. As for Black-Box MOO, these methods are computationally expensive and often suboptimal within finite optimization steps, generally less effective than White-Box MOO methods.

## 3 PRELIMINARIES

### 3.1 PARETO OPTIMAL

In this work, we consider $m$ tasks described by $f(\boldsymbol{x}) := [f_i(\boldsymbol{x})] : \mathbb{R}^n \to \mathbb{R}^m$ for any point $\boldsymbol{x}$ in *Solution Space* $\mathbb{R}^n$, where each $f_i(\boldsymbol{x}) : \mathbb{R}^n \to \mathbb{R}, i \in [m]$ represents the performance of the $i$-th task to be maximized. Given an desired target $\boldsymbol{y} \in \mathbb{R}^m$, we set a non-negative objective function $\mathcal{L}(f(\boldsymbol{x}), \boldsymbol{y}) = [l_1, \ldots, l_m]^\mathsf{T} : \mathbb{R}^m \to \mathbb{O}^m$ to be a non-negative objective function mapping the *Value Space* $\mathbb{R}^m$ to the *Objective Space* $\mathbb{O}^m$, where $l_i$ for $i \in [m]$ is the objective function of the $i$-th task. Hence, maximizing the performance $f(\boldsymbol{x})$ is equivalent to minimizing the objective function. Consequently, we have $l_i^{\boldsymbol{x}'} - l_i^{\boldsymbol{x}} \geq 0$ if $f_i(\boldsymbol{x}') \leq f_i(\boldsymbol{x})$ for two points $\boldsymbol{x}, \boldsymbol{x}' \in \mathbb{R}^n$.

For any two points $\boldsymbol{x}, \boldsymbol{x}' \in \mathbb{R}^n$, $\boldsymbol{x}$ dominates $\boldsymbol{x}'$, denoted by $\mathcal{L}^{\boldsymbol{x}'} \succeq \mathcal{L}^{\boldsymbol{x}}$, if and only if $\mathcal{L}^{\boldsymbol{x}'} - \mathcal{L}^{\boldsymbol{x}} \in \mathbb{R}_+^m$, where $\mathbb{R}_+^m := \{\mathcal{L} \in \mathbb{O}^m | l_i \geq 0, \forall i \in [m]\}$. The partial ordering $\mathcal{L}^{\boldsymbol{x}'} \succeq \mathcal{L}^{\boldsymbol{x}}$ implies $l_i^{\boldsymbol{x}'} - l_i^{\boldsymbol{x}} \geq 0, \forall i \in [m]$. When $\boldsymbol{x}$ strictly dominates $\boldsymbol{x}'$, denoted by $\mathcal{L}^{\boldsymbol{x}'} \succ \mathcal{L}^{\boldsymbol{x}}$, it means there is at least one $i$ for which $l_i^{\boldsymbol{x}'} - l_i^{\boldsymbol{x}} > 0$. Geometrically, $\mathcal{L}^{\boldsymbol{x}'} \succ \mathcal{L}^{\boldsymbol{x}}$ means that $\mathcal{L}^{\boldsymbol{x}'}$ lies in the positive cone

pivoted at $\mathcal{L}^{\boldsymbol{x}}$, i.e. $\mathcal{L}^{\boldsymbol{x}'} \in \{\mathcal{L}^{\boldsymbol{x}}\} + \mathbb{R}_+^m := \{\mathcal{L}^{\boldsymbol{x}} + \mathcal{L} \mid \mathcal{L} \in \mathbb{R}_+^m\}$. A point $\boldsymbol{x}^*$ is said to be **Pareto optimal** if $\boldsymbol{x}^*$ is not dominated by any other points in $\mathbb{R}^n$. Similarly, $\boldsymbol{x}^*$ is locally Pareto optimal if $\boldsymbol{x}^*$ is not dominated by any other points in the neighborhood of $\boldsymbol{x}^*$, i.e. $\mathcal{N}(\boldsymbol{x}^*)$. The set of all Pareto optimal solutions is defined as:

$$\mathcal{P} := \left\{\boldsymbol{x}^* \in \mathbb{R}^n \mid \forall \boldsymbol{x} \in \mathbb{R}^n - \{\boldsymbol{x}^*\}, \mathcal{L}^{\boldsymbol{x}^*} \not\succeq \mathcal{L}^{\boldsymbol{x}}\right\}, \tag{1}$$

where $\mathcal{L}^{\boldsymbol{x}^*} \not\succeq \mathcal{L}^{\boldsymbol{x}}$ represents $\boldsymbol{x}^*$ is not dominated by other point $\boldsymbol{x}$. The set of multi-objective values of the Pareto optimal solutions is called **Pareto front**:

$$\mathcal{F} := \left\{\mathcal{L}^{\boldsymbol{x}^*} \mid \boldsymbol{x}^* \in \mathcal{P}\right\}. \tag{2}$$

## 3.2 Non-Uniformity

The function termed as **Non-Uniformity** (Mahapatra & Rajan, 2020) quantitatively evaluates the degree of misalignment between the objective value $\mathcal{L}$ and the preference vector $\lambda \in \mathbb{R}^m$:

**Definition 3.1** (Non-Uniformity). *For any point $\boldsymbol{x} \in \mathbb{R}^n$, the Non-Uniformity of its objective values $\mathcal{L}$ in relation to a given preference vector $\lambda \in \mathbb{R}^m$ as:*

$$\mu_\lambda(\mathcal{L}) = \sum_{i=1}^m \hat{l}_i \log\left(\frac{\hat{l}_i}{1/m}\right) = \mathrm{KL}\left(\hat{\mathcal{L}} \mid \frac{\mathbf{1}}{m}\right), \tag{3}$$

*where $\hat{\mathcal{L}} = [\hat{l}_1, \ldots, \hat{l}_m]^\mathsf{T}$ and $\hat{l}_i$ is the weighted normalization $\hat{l}_i = \frac{\lambda_i l_i}{\sum_{i'=1}^m \lambda_{i'} l_{i'}}$.*

The Kullback-Leibler (KL) divergence of $\hat{\mathcal{L}}$ from the uniform distribution $\frac{\mathbf{1}}{m}$ characterizes non-uniformity. When the objective value fulfills the preference condition, we have $\mu_\lambda(\mathcal{L}) = 0$; otherwise, $\mu_\lambda(\mathcal{L}) > 0$. Consequently, we prefer a lower $\mu_\lambda(\mathcal{L})$.

## 4 Method

In this study, we explore a **Preference-Conditioned Inversion** (PCI) framework catering to given drug design requirements, which allows for generating diverse molecules at any property ratio. We illustrate the pipeline in Figure. 2, then describe the key steps following the order:

- **Differentiable Surrogate Oracle.** We construct differentiable surrogate property functions (also known as oracle) to be maximized in molecular optimization (Section 4.2). Surrogate oracle is pre-trained once and *freezed* in the following steps.
- **Preference Guided Pareto Molecular Opimization.** We formulate the discrete molecule optimization into a locally differentiable problem. Then we can be optimize the molecules with the non-dominating gradients (Section 4.3).

## 4.1 Problem Formulation

**Definition 4.1** (MOMO). *Given the Chemical Space $\mathcal{X}$, oracle function $f(\boldsymbol{x})$, objective function $\mathcal{L}$, the target property score $\boldsymbol{y} \in \mathbb{R}^m$, multi-objective molecule optimization is to find candidate molecules $\boldsymbol{x}^* \in \mathcal{X}$ that simultaneously minimize all objective functions:*

$$\boldsymbol{x}^* = \arg\min_{\boldsymbol{x} \in \mathcal{X}} \mathcal{L}(f(\boldsymbol{x}), \boldsymbol{y}). \tag{4}$$

In other words, MOMO's goal is to find the set $\mathcal{P}$ of all Pareto optimal molecules, which is challenging in practice (Miettinen, 1999; Ehrgott, 2005). Therefore, we focus on a subproblem of MOMO, defined as **Preference-Conditioned MOMO** to find Pareto optimal molecules satisfying the given preference condition.

**Definition 4.2** (Preference-Conditioned MOMO). *Given the Chemical Space $\mathcal{X}$, oracle function $f(\boldsymbol{x})$, objective function $\mathcal{L}$, the target property score $\boldsymbol{y} \in \mathbb{R}^m$, preference vector $\lambda \in \mathbb{R}^m$, the preference-conditioned multi-objective molecular optimization is to find molecules that satisfying:*

$$\boldsymbol{x}^* \in \mathcal{P}_\lambda = \left\{\boldsymbol{x}^* \in \mathcal{P} \mid \lambda_1 l_1^* = \cdots = \lambda_j l_j^* = \cdots = \lambda_m l_m^*\right\}. \tag{5}$$

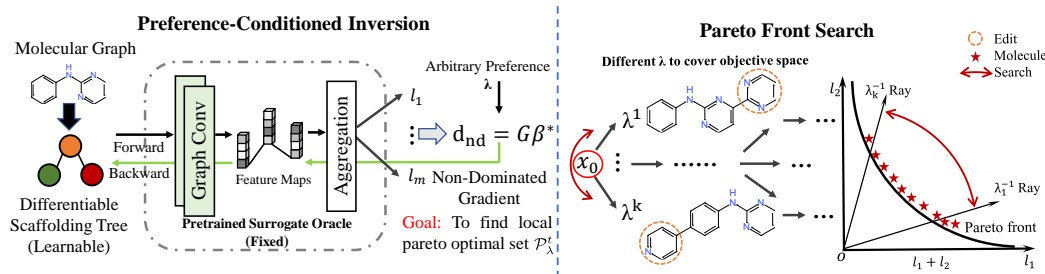

Figure 2: (1) **Preference-Conditioned Inversion (PCI)**. Initially, a differentiable surrogate oracle model is trained and then fixed for Inversion. In each iteration, PCI is able to find local preference-conditioned Pareto-optimal solutions $\mathcal{P}_\lambda^t$ under the guidance of non-dominated gradient. (2) **Pareto Front Search**. Exploring the Pareto front approximately with different preferences, thereby assisting in the discovery of diverse molecules with varying property ratios.

Preference-Conditioned MOMO's goal is to find the set $\mathcal{P}_\lambda$ of all Pareto optimal molecules that satisfying additional preference condition. Naturally, it follows that $\mathcal{P}_\lambda \subseteq \mathcal{P}$. Therefore, we can approximately obtain the whole Pareto optimal set by $\mathcal{P} = \bigcup_\lambda \mathcal{P}_\lambda$.

## 4.2 DIFFERENTIABLE SURROGATE ORACLE

In this section, we aims to develop a differentiable Surrogate Oracle model to capture knowledge from the ground truth oracle $\mathcal{O}$. We imitate the oracle function $\mathcal{O}$ using a multi-head architecture to independently predict $m$ property scores $\widehat{\boldsymbol{y}} \in \mathbb{R}^m$:

$$\widehat{\boldsymbol{y}} = f(\boldsymbol{x}; \theta) \approx [\mathcal{O}_1(\boldsymbol{x}), \mathcal{O}_2(\boldsymbol{x}), \cdots, \mathcal{O}_m(\boldsymbol{x})]^\mathsf{T} = \boldsymbol{y}, \tag{6}$$

where $\theta$ is the learnable parameters. We train the model by minimizing the discrepancy between the prediction $\widehat{\boldsymbol{y}}$ and the ground truth $\boldsymbol{y}$:

$$\theta^* = \arg \min_\theta \mathcal{L}\left(\boldsymbol{y} = [\mathcal{O}_1(\boldsymbol{x}), \mathcal{O}_2(\boldsymbol{x}), \cdots, \mathcal{O}_m(\boldsymbol{x})]^\mathsf{T}, \widehat{\boldsymbol{y}} = f(\boldsymbol{x}; \theta)\right), \tag{7}$$

where $\mathcal{L}$ is loss function, e.g. binary cross entropy. The parameters of the pretrained surrogate oracle model are *freezed* in the following molecular optimization stage. In MOMO task, we adopt differential scaffolding tree $\widetilde{\mathcal{T}}_{\boldsymbol{x}}$ (Fu et al., 2022) as molecular $\boldsymbol{x}$'s representation, which contains learnable parameters in the tree structure. It allows us to search the neighborhood set $\mathcal{N}(\boldsymbol{x})$ via node shrinking, replacement, or expansion.

## 4.3 PREFERENCE GUIDED PARETO MOLECULAR OPTIMIZATION

In this section, we introduce an approach to find Pareto optimal molecules over preference requirement. Inspired by Fu et al. (2022), we formulate the discrete molecule optimization into a locally differentiable problem. At the $t$-th iteration, given one molecular $\boldsymbol{x}^t$, PCI aims to find local Pareto optimal molecules set $\mathcal{P}_\lambda^t$ over preference condition $\lambda$ from neighborhood set $\mathcal{N}(\boldsymbol{x}^t)$.

$$\boldsymbol{x}^* \in \mathcal{P}_\lambda \approx \boldsymbol{x}^{(t+1)} \in \mathcal{P}_\lambda^t \subseteq \mathcal{N}(\boldsymbol{x}^t) \tag{8}$$

**Identifying $\mathcal{P}_\lambda^t$ with Non-Dominating Descent Direction.** Let $g_i = \nabla l_i$ represent the gradient of the $i$-th property objective function. Consequently, we obtain $G = \nabla \mathcal{L} = [g_1, \ldots, g_m]$. To approach the Pareto front, Désidéri et al. Désidéri (2012) demonstrated that the descent direction $d$ can be found within the convex hull of the gradients, i.e., $d \in \mathcal{CH}_\theta := \{G\beta\}$, where $\beta \in \mathcal{S}^m$ belongs to the $m$-dimensional simplex. In order to identify a non-dominating descent direction, $d_{nd} = G\beta^*$, which aligns with the preference vector while continuing to move towards the Pareto front, we follow Mahapatra & Rajan (2020) and solve the following Linear Programming (LP) problem at the $k$-th Inversion steps:

$$\beta^* = \arg \max_{\beta \in \mathcal{S}^m} \beta^T C \left(a \left(1 - \mathbb{I}_\mu^k\right) + \mathbf{1}\mathbb{I}_\mu^k\right)$$
$$\text{s.t.} \quad \beta^T c_j \geqslant a^T c_j \mathbb{I}_J, \qquad\qquad j \in \bar{J} - J^*, \tag{9}$$
$$\beta^T c_j \geqslant 0, \qquad\qquad j \in J^*,$$

---

**Algorithm 1:** Preference-Conditioned Inversion (PCI)

---

**Input:** Input molecule $\boldsymbol{x}^0 \in \mathcal{X}$, preference vector $\lambda \in \mathbb{R}^m$, and step size $\eta > 0$.
**Output:** Generated Molecule $\boldsymbol{x}^*$.

1 Initialization.
2 Train surrogate oracle according to Eq. 7.
3 **for** $t = 0, \ldots, T$ **do**
4      Convert molecule $\boldsymbol{x}^t$ to differentiable scaffolding tree $\widetilde{\mathcal{T}}_{\boldsymbol{x}^t}^0$;
5      **for** $k = 0, \ldots, K$ **do**
6          Compute gradients of each property objective w.r.t. $\widetilde{\mathcal{T}}_{\boldsymbol{x}^t}^k$: $G = \nabla\mathcal{L} = [g_1, \ldots, g_m]$;
7          Determine $\beta^*$ by solving the Linear Programming (LP) problem as Eq. 9;
8          Calculate the non-dominating direction of descent $d_{nd} = G\beta^*$;
9          Update the differentiable scaffolding tree using $\widetilde{\mathcal{T}}_{\boldsymbol{x}^t}^{k+1} = \widetilde{\mathcal{T}}_{\boldsymbol{x}^t}^k - \eta d_{nd}$.
10      **end**
11      Sample discrete $\mathcal{T}_{\boldsymbol{x}^{t+1}}$ from continuous $\widetilde{\mathcal{T}}_{\boldsymbol{x}^t}^K$ and assemble it to molecule $\boldsymbol{x}^{t+1}$.
12 **end**

---

where $\mathbb{I}_\mu^k$ is an indicator for non-zero Non-Uniformity, $\mu_\lambda^k$; $C = G^\top G \in \mathbb{R}^{m \times m}$ is a symmetric matrix with $c_j$ as its columns; $J^* = \left\{ j \mid \lambda_j l_j^k = \max_{j'}\left\{\lambda_{j'} l_{j'}^k\right\} \right\}$ represents the index of the maximum relative objective values; $\bar{J}$ is the index set for the gradients that ascends during the optimization step; $\mathbb{I}_J$ is an indicator for not all objectives ascending simultaneously, i.e. $\bar{J} \neq [m]$; and $a$ is the adjustment, and $a_j = \lambda_j \left(\log\left(\frac{\hat{i}_i}{1/m}\right) - \mu_\lambda^k\right)$. Finally, the non-dominating direction $d_{nd} = G\beta^*$ is acquired, and we update the differentiable scaffolding tree by $\widetilde{\mathcal{T}}_{\boldsymbol{x}} = \widetilde{\mathcal{T}}_{\boldsymbol{x}} - \eta d_{nd}$.

**Preference-Conditioned Inversion.** At the $t$-th iteration, we begin with a molecule $\boldsymbol{x}^t$ and convert it to its corresponding differentiable scaffolding tree $\widetilde{\mathcal{T}}_{\boldsymbol{x}^t}$. Subsequently, we identify the preference-conditioned local Pareto optimal solution $\widetilde{\mathcal{T}}_{\boldsymbol{x}^t}^*$ within the neighborhood set $\mathcal{N}(\mathcal{T}_{\boldsymbol{x}^t})$ by performing $K$ rounds of gradient descent against the non-dominating direction $d_{nd}$. From $\widetilde{\mathcal{T}}_{\boldsymbol{x}^t}^*$, we can sample the discrete scaffolding tree $\mathcal{T}_{\boldsymbol{x}^t}^*$ and assemble it to molecules, denoted as $\boldsymbol{x}^{t+1}$ in the following iteration. Our proposed Preference-Conditioned Inversion (PCI) framework is summarized in Algorithm 1. The time complexity is summarized in Appendix Sec. B.

**Convergence of PCI towards Preference-Conditioned Solutions in Discrete Chemical Space.**
We present the theoretical analysis of the Preference-Conditioned Inversion (PCI) Algorithm, discussing its convergence properties. The PCI algorithm guarantees convergence towards preference-conditioned solutions in discrete chemical space, providing optimal uniformity and non-dominating properties. The proof and detailed explanation can be found in Appendix Sec. A.

**Theorem 4.3** (Approximation Guarantee). *Under the assumptions stated in Sec. A, when the PCI algorithm is applied with an initial molecule $\boldsymbol{x}^0$ and preference vector $\lambda$, it guarantees the following approximation when performing $T$ optimization rounds:*

$$\mathcal{L}^T \in \mathcal{A} := \left\{\mathcal{L} \in \mathcal{O} \mid \mathcal{L} \preceq (\Gamma\check{\lambda}^* + (1-\Gamma)\check{\lambda}^0) \cdot \lambda^{-1}\right\}, \tag{10}$$

*where $\Gamma = \frac{1-\alpha^T}{(1-\alpha)N}$, $\lambda^{-1} := (1/\lambda_i, \ldots, 1/\lambda_m)$, $\check{\lambda}^*$ and $\check{\lambda}^0$ is the maximum relative objective value $\check{\lambda}^t := \max\left\{\mathcal{L}_j^t \lambda_j \mid j \in [m]\right\}$ of $\boldsymbol{x}^*$ and $\boldsymbol{x}^0$.*

Theorem 4.3 demonstrates the convergence of PCI towards the preference-conditioned Pareto optimal solution $\boldsymbol{x}^*$, which exhibits optimal uniformity and non-dominating properties. The maximum relative objective value $\check{\lambda}^t$ is associated with the upper bound of the admissible set. Therefore, a lower $\check{\lambda}^t$ indicates that we are closer to the preference-conditioned Pareto optimal solution, resulting in improved uniformity and higher properties. At the $T$-th step, the maximum distance between the molecule $\boldsymbol{x}^T$ we obtained and the optimal solution $\boldsymbol{x}^*$ is bounded, i.e. $\|\check{\lambda}^0 - \check{\lambda}^T\| \geq \frac{1-\alpha^T}{(1-\alpha)N}\|\check{\lambda}^0 - \check{\lambda}^*\|$.

## 5 EXPERIMENTS

### 5.1 SYNTHETIC TASK

This section demonstrates the performance of our proposed algorithm, utilizing a synthetic objective from Lin et al. (2019). We aim to minimize two non-convex objective functions, denoted as:

$$l_1(\boldsymbol{x}) = 1 - e^{-\left\|\boldsymbol{x} - \frac{1}{\sqrt{n}}\right\|_2^2}, \ \ l_2(\boldsymbol{x}) = 1 - e^{-\left\|\boldsymbol{x} + \frac{1}{\sqrt{n}}\right\|_2^2}, \tag{11}$$

where $\boldsymbol{x}$ represents a point in discrete space, with its dimension set to $n = 20$. Since we have access to the details of objective functions, we can obtain the ground truth of the Pareto front.

**Implementation Details.** All methods used the same 5 uniformly distributed preference vectors. PCI and Linear Scalarization were optimized from scratch for each preference. NSGA-III was used in the Evolutionary Algorithms, with preference vectors serving as the reference point. The process involved 300 iterations, a population size of 40, and 10 offspring. The Multi-Objective Bayesian Optimization utilized the expected hypervolume improvement as an acquisition function, initializing 1000 points and executing 100 optimization loops.

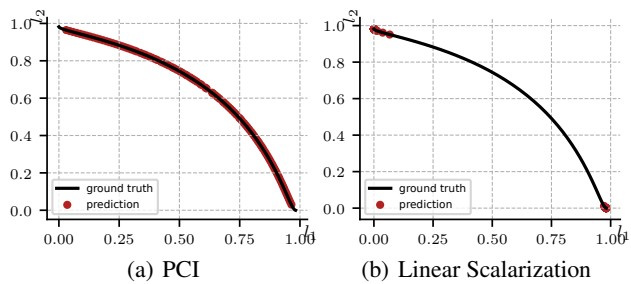

(a) PCI      (b) Linear Scalarization

Figure 3: Ground truth Pareto front (black solid curve) and solutions (red circles) obtained by (a) PCI: can explore almost the entire Pareto front; (b) Linear Scalarization: tends to cluster at the ends.

**Experimental Results.** Figure. 1 illustrates that our PCI framework not only captures specific solutions based on given preferences but also but also achieves the highest efficiency. For evolutionay algorithm, we only show the non-dominated solutions in the $n$-th generation. For PCI and Linear Scalarization, which incorporate preference information, we also utilized preferences to scan the entire Pareto front, as shown in Figure. 3. Our proposed PCI can explore almost the entire Pareto front. In contrast, the solutions of Linear Scalarization tend to cluster at the ends of the Pareto front, leaving certain regions unexplored.

### 5.2 MULTI-OBJECTIVE MOLECULAR OPTIMIZATION.

In this section, we first demonstrate that PCI's capacity for generating molecules over given preference vectors. Subsequently, we use different preferences to evaluate the effectiveness of PCI in synthesizing diverse molecules. More details please refer to Appendix Sec. C.

**Dataset.** We train the surrogate model on the ZINC 250K dataset (Sterling & Irwin, 2015), which consists of approximately 250K drug-like molecules extracted from the ZINC database. In accordance with DST (Fu et al., 2022), we select the substructures that appear more than 1000 times as the vocabulary set $S$, which consists of 82 frequent substructures. We remove molecules containing out-of-vocabulary substructures, resulting in a remaining dataset of 195K molecules.

**Properties and Oracles.** (1) QED ranging in $[0, 1]$ that provides a quantitative assessment of a molecule's drug-likeness. (2) SA evaluates the ease of synthesizing a molecule, and is normalized to $[0, 1]$ (Gao & Coley, 2020). (3) JNK3 is a member of the mitogen-activated protein kinase family, with scores ranging in $[0, 1]$. (4) GSK3$\beta$ is an enzyme encoded by the GSK3$\beta$ gene in humans, and also has a range of $[0, 1]$. For all aforementioned properties, higher values indicate better performance. To evaluate QED and SA, we utilize the RDKit package. Furthermore, we employ random forest to evaluate GSK3$\beta$ and JNK3, as described by Li et al. (2018) and Jin et al. (2020).

**Metric.** (1)Novelty (Nov) represents the proportion of generated molecules not in the training set. (2) Top-K Diversity (Div) (Bengio et al., 2021; Fu et al., 2022) of generated molecules is defined as the average pairwise Tanimoto distance between the Morgan fingerprints. (3) Top-K Average Property Score (APS) (Bengio et al., 2021; Fu et al., 2022) refers to the average score of the top-

100 molecules. (5) Non-Uniformity evaluates the extent of misalignment between properties and preferences vector.

**Preference Guided Molecular Optimization.** In this task, our goal is to evaluate that if PCI can generate molecules over given preference. Here we select two properties, GSK3$\beta$ and JNK3 and the same 5 uniformly distributed preference vectors to serve as independent trials. **I-LS** is a baseline that using the same

Table 1: Preference Guided Molecular Optimization.

| Method | Nov↑ | Div↑ | APS↑ | Non-Uniformity($\mu$)↓ |
|---|---|---|---|---|
| MOEA/D | **100%** | n/a | 0.287 | 0.083 |
| NSGA-III | **100%** | n/a | 0.342 | 0.112 |
| I-LS | **100%** | **0.53** | 0.543 | 0.051 |
| PCI | **100%** | 0.42 | **0.659** | **0.026** |

Inversion framework with PCI but adopting Linear Scalarization. For MOEA/D and NSGA-III, we report the performance of the molecule with lowest Non-Uniformity. For I-LS and PCI ,we calculate the performance of the top-20 molecules in terms of Non-Uniformity per preference. For each preference, we compute the results separately and report the average results across all five trials. The results are shown in Table. 1. PCI outperforms all most baselines by a significant margin. Our proposed PCI achieves better uniformity and higher APS. We find that the diversity of PCI is lower than that of I-LS. A reasonable explanation is that the molecules produced by PCI concentrate more around preference with a lower diversity. This superior performance is attributed to our method's ability to identify solutions that are more specifically related to the preference vector.

**Multi-Objective de novo Design.** In this task, we use different preferences to evaluate the effectiveness of PCI in synthesizing diverse molecules. For I-LS and PCI, we collected all the solutions generated by varying preferences to report the final results. The results are detailed in Table 2. Here oracle named "$A + B$" means we allocate $A$ oracle call budget for surrogate oracle pretraining, and $B$ for Inversion. PCI exhibits superior performance compared to the majority of baselines. Diversity and APS is a common trade-off. Some methods encounter difficulties in simultaneously achieving high diversity scores and APS, due to their limited capacity to explore the chemical space. Nevertheless, our approach manifests robust performance on both metrics.

Table 2: Multi-Objective de novo Design.

| Method | GSK3$\beta$ + JNK3 | | | | GSK3$\beta$+JNK3+QED+SA | | | |
|---|---|---|---|---|---|---|---|---|
| | Nov↑ | Div↑ | APS↑ | oracle↓ | Nov↑ | Div↑ | APS↑ | oracle↓ |
| LigGPT | **100%** | **0.845** | 0.271 | 100K+0 | **100%** | **0.902** | 0.378 | 100K+0 |
| GCPN | **100%** | 0.578 | 0.293 | 0+200K | **100%** | 0.596 | 0.450 | 0+200K |
| MolDQN | **100%** | 0.605 | 0.348 | 0+200K | **100%** | 0.597 | 0.365 | 0+200K |
| GA+D | **100%** | 0.657 | 0.608 | 0+50K | 97% | 0.681 | 0.632 | 0+50K |
| RationaleRL | **100%** | 0.700 | 0.795 | 25K+67K | 99% | 0.720 | 0.675 | 25K+67K |
| MARS | **100%** | 0.711 | 0.789 | 0+50K | **100%** | 0.714 | 0.662 | 0+50K |
| ChemBO | 98% | 0.702 | 0.747 | 0+50K | 99% | 0.701 | 0.648 | 0+50K |
| BOSS | 99% | 0.564 | 0.504 | 0+50K | 98% | 0.561 | 0.504 | 0+50K |
| LSTM | **100%** | 0.712 | 0.680 | 0+50K | **100%** | 0.706 | 0.672 | 0+50K |
| Graph-GA | **100%** | 0.634 | 0.825 | 0+25K | **100%** | 0.723 | 0.714 | 0+25K |
| DST | **100%** | 0.750 | 0.827 | 10K+5K | **100%** | 0.755 | 0.752 | 20K+5K |
| MOGFN-PC | **100%** | 0.673 | 0.742 | 50K+20K | **100%** | 0.711 | 0.621 | 50K+20K |
| RetMol | **100%** | 0.688 | 0.769 | 50K+5K | **100%** | 0.691 | 0.642 | 50K+5K |
| I-LS | **100%** | 0.693 | 0.823 | 10K+5K | **100%** | 0.704 | 0.734 | 20K+5K |
| PCI | **100%** | 0.768 | **0.841** | 10K+5K | **100%** | 0.769 | **0.773** | 20K+5K |

## 5.3 ABLATION STUDY

**Case Study.** We select an initial molecule $x_0$ and a preference vector $[0.25, 0.75]$. To facilitate understanding, we focus on two properties, JNK3 and GSK3$\beta$. In each step, we greedily add one substructure for PCI and I-LS. We demonstrate the optimization process in Figure. 4(a). Also, we display the corresponding molecular graphs, property scores, and calculate the loss ratio: $ratio = \frac{l_{JNK3}}{l_{GSK3\beta}}$ in Figure. 4(b). It demonstrates that PCI is capable of finding preference-specific Pareto optimal solutions.

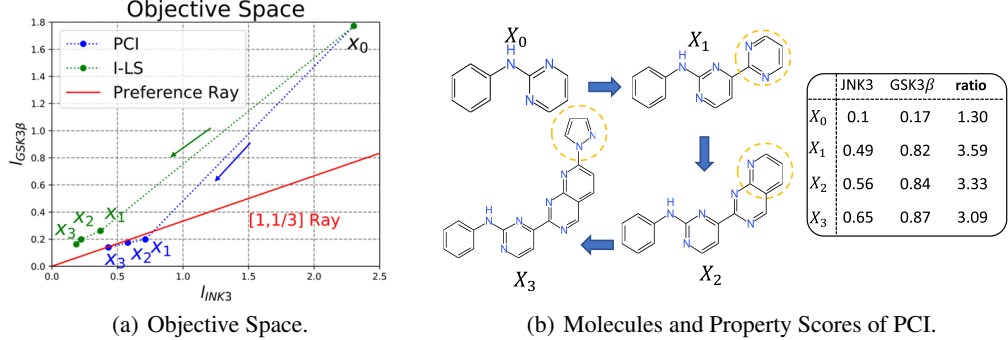

(a) Objective Space.  (b) Molecules and Property Scores of PCI.

Figure 4: Optimization process of PCI and I-LS. (a) Loss trend and corresponding preference ray. (b) Generated molecules by PCI, property scores and loss ratio.

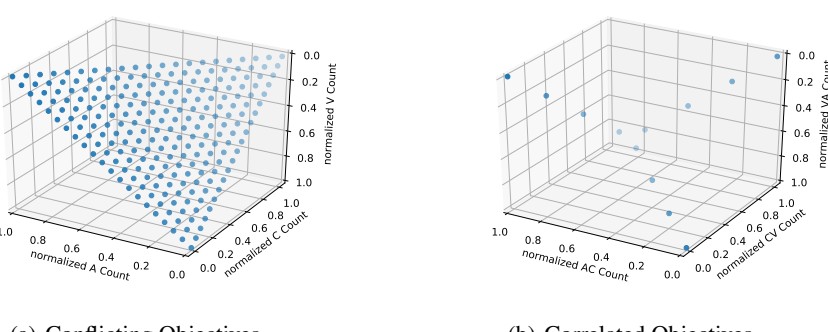

(a) Conflicting Objectives.  (b) Correlated Objectives.

Figure 5: Optimization process of PCI and I-LS. (a) Loss trend and corresponding preference ray. (b) Generated molecules by PCI, property scores and loss ratio.

**Search Efficiency.** To understand the search efficiency of PCI, we search the Pareto front for MOMO under a limited budget for the number of preference vectors $(2, 5, 10, 15, 20)$. For GNK3$\beta$+JNK3, we allocate a $10K$ oracle call budget for surrogate oracle pretraining, and $N_{preference} * 1K$ for Inversion. For the optimization involving GNK3$\beta$+JNK3+QED+SA, the pretraining budget is fixed at $20K$. Figure. 6 clearly illustrates that APS increases rapidly with number of preference.

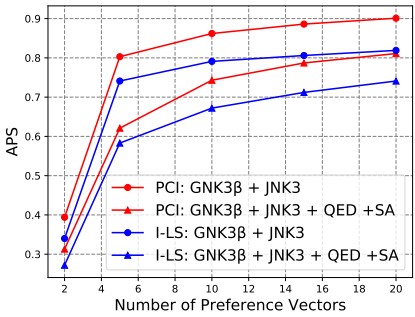

Figure 6: Search Efficiency. The number of preferences represents the scope that we search the Pareto front. The number of oracle calls grows with the number of preferences.

**Conflicting and Correlated Objectives.** For fair comparison, we follow Jain et al. (2023) and use the synthetic sequence design task from Stanton et al. (2022). The task consists of generating strings with the objectives given by occurrences of a set of d n-grams. PCI adequately models the trade-off between conflicting objectives in the 3 Unigrams task as illustrated by the Pareto front of generated candidates in Figure.5(a). For the 3 Bigrams task with correlated objectives, Figure. 5(b) demonstrates PCI generates candidates which can simultaneously maximize multiple objectives.

## 6 CONCLUSION

In this work, we present a novel approach, preference-conditioned Inversion (PCI) framework, to explore potential trade-offs between multiple properties for molecular optimization. PCI combines the strengths of Inversion models and Pareto optimization to generate high-quality, diverse molecules that satisfy preference constraints. Comprehensive experimental evaluations reveal that our model is able to select molecules across various preferences and significantly outperform existing baselines.

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

# A    THEORETICAL ANALYSIS

In this section, we discuss the theoretical properties of Preference-Conditioned Inversion (PCI) Algorithm.

## A.1    ASSUMPTIONS AND KEY LEMMAS

**Assumption A.1** (Molecule Size Bound). *The sizes (i.e., number of substructures) of all the scaffolding trees generated are upper bounded by $N$.*

We focus on small molecule optimization; the target molecular properties would decrease significantly when the molecule size is too large (Bickerton et al., 2012), e.g., QED. To perform a convergence analysis, we initially establish several assumptions to characterize the geometry of the objective landscape.

**Assumption A.2** (Submodularity and Smoothness). *Suppose $x_{t-1}, x_t, x_{t+1}$ are generated successively by PCI via growing a substructure on the scaffolding tree. We assume that the corresponding objective gain (i.e., $\triangle \check{\lambda}^t$) satisfies the diminishing returns property:*

$$\check{\lambda}^{t-1} - \check{\lambda}^t \geq \check{\lambda}^t - \check{\lambda}^{t+1}, \quad (submodularity) \tag{12}$$

*Submodularity plays the role of concavity/convexity in the discrete regime. On the other hand, we specify the curvature ratio of the objective function $\mathcal{L}$ by assuming*

$$\check{\lambda}^t - \check{\lambda}^{t+1} \geq \alpha(\check{\lambda}^{t-1} - \check{\lambda}^t), \quad 0 < \alpha < 1. \quad (curvature) \tag{13}$$

The choice of submodularity as an assumption for our analysis was motivated by experimental observations. For instance, in the optimization process, we noticed that the objective values, such as QED, can rapidly increase to a high point with just a few iterations. However, as we added more atoms, the growth rate began to decrease, i.e. the return is diminishing when the property scores are reaching the upper bound. This trend is observed in many properties and provides insight into our assumption.

**Definition A.3** (Dominant Set). *Given $x^t$ in chemical space $\mathcal{X}$ at the $t$-th iteration, we define a Dominant Set $\mathcal{V}_{\preceq \mathcal{L}^t} \subset \mathbb{R}^m$ that contains all attainable multi-objective values that dominate the $\mathcal{L}^t$ as:*

$$\mathcal{V}_{\preceq \mathcal{L}^t} = \left\{ \mathcal{L} \in \mathcal{O} \mid \mathcal{L} \preceq \mathcal{L}^t \right\}. \tag{14}$$

**Definition A.4** (Uniform Set). *Given a molecule $x^t$ in the chemical space $\mathcal{X}$ at the $t$-th iteration and a specified preference vector $\lambda \in \mathbb{R}^m$, we define a Uniform Set $\mathcal{M}_{\mathcal{L}^t}^\lambda \subset \mathbb{R}^m$ that contains all attainable multi-objective values demonstrating enhanced uniformity compared to $\mathcal{L}^t$ as:*

$$\mathcal{M}_{\mathcal{L}^t}^\lambda = \left\{ \mu_\lambda(\mathcal{L}) \leq \mu_\lambda(\mathcal{L}^t) \right\}. \tag{15}$$

**Definition A.5** (Admissible Set). *Given a molecule $x^t$ in the chemical space $\mathcal{X}$ at the $t$-th iteration and a specified preference vector $\lambda \in \mathbb{R}^m$, we define a bounded Admissible Set $\mathcal{A}_{\mathcal{L}^t}^\lambda \subset \mathbb{R}^m$ as:*

$$\mathcal{A}_{\mathcal{L}^t}^\lambda = \left\{ \mathcal{L} \in \mathcal{O} \mid \mathcal{L} \preceq \check{\mathcal{L}}^t \right\}, \tag{16}$$

*where $\check{\mathcal{L}}^t = \check{\lambda}^t (1/\lambda_1, \cdots, 1/\lambda_m)$, and $\check{\lambda}^t = \max \left\{ \mathcal{L}_j^t \lambda_j \mid j \in [m] \right\}$.*

Clearly, the admissible set contains all the points in $\mathcal{O}$ that dominate the $\mathcal{L}^t$, i.e. $\mathcal{V}_{\preceq \mathcal{L}^t} \subset \mathcal{A}_{\mathcal{L}^t}^\lambda$. Moreover, when $\mu_\lambda(\mathcal{L}) > 0$, it also contains points exhibiting superior uniformity compared to $\check{\mathcal{L}}^t$, i.e. $\mathcal{A}_{\mathcal{L}^t}^\lambda \cap \mathcal{M}_{\mathcal{L}^t}^\lambda \neq \emptyset$. Consequently, the admissible set encompasses the desired solution for the subsequent iteration, fulfilling both uniformity and dominating properties.

As illustrated in Mahapatra & Rajan (2020), a descent direction $d_{nd}$ will be oriented towards the preference-specific Pareto front and within a confined admissible set. Since we perform $K$ descents in each iteration, we restructure properties for PCI in the molecular optimization problem as follows:

**Lemma A.6** (Bounded Objective Space for the Next Iteration). *There exists a step size $\eta_0 > 0$, such that for every $\eta \in [0, \eta_0]$, PCI employs $\widetilde{\mathcal{T}}_{x^t} = \widetilde{\mathcal{T}}_{x^t} - \eta d_{nd}$ to update differentiable scaffolding tree until convergence. Subsequently, we greedily sample a molecule as $x^{t+1}$ from $\widetilde{\mathcal{T}}_{x^t}^K$ by adding a substructure, if the solution is nonempty. The multi-objective value $\mathcal{L}^{t+1}$ of the new solution point $x^{t+1}$ lies in the $t$-th admissible set:*

$$\mathcal{L}^{t+1} \in \mathcal{A}_{\mathcal{L}^t}^\lambda. \tag{17}$$

Following Theorem 2 in Mahapatra & Rajan (2020), the empirical loss $\mathcal{L}_{\widetilde{\mathcal{T}}}$ of scaffolding tree $\widetilde{\mathcal{T}}_{\boldsymbol{x}^t}^K$ lies in the t-th admissible set $\mathcal{A}_{\mathcal{L}^t}^\lambda$. Since we greedily sample a molecule as $\boldsymbol{x}^{t+1}$ from $\widetilde{\mathcal{T}}_{\boldsymbol{x}^t}^K$, thus we have $\mathcal{L}^{t+1} \preceq \mathcal{L}_{\widetilde{\mathcal{T}}}$. Therefore, it follows that $\mathcal{L}^{t+1} \in \mathcal{A}_{\mathcal{L}^t}^\lambda$. It demonstrates that for a molecule $\boldsymbol{x}^t$ at the t-th iteration, PCI selects a molecule as $\boldsymbol{x}^{t+1}$ from $\boldsymbol{x}^t$'s neighborhood set $\mathcal{N}(\boldsymbol{x}^t)$, moving towards improved uniformity and dominating properties. It provides a theoretical guarantee for the quality of the solution.

**Corollary A.7** (Convergence of Admissible Set). *The sequence of relative maximum values $\check{\lambda}^t$ obtained by descending against the adjusted gradient $d_{nd}$ is monotonic with $\check{\lambda}^{t+1} \leq \check{\lambda}^t$, which means*

$$\mathcal{A}_{\mathcal{L}^t}^\lambda \subset \mathcal{A}_{\mathcal{L}^{t+1}}^\lambda, \tag{18}$$

*and the sequence of bounded sets $\{\mathcal{A}_{\mathcal{L}^{t+1}}^\lambda\}$ converges.*

Since $\mathcal{L}^{t+1} \in \mathcal{A}_{\mathcal{L}^t}^\lambda$, we naturally get $\check{\lambda}^{t+1} \leq \check{\lambda}^t$, thus we have $\mathcal{A}_{\mathcal{L}^t}^\lambda \subset \mathcal{A}_{\mathcal{L}^{t+1}}^\lambda$. It demonstrates the monotonicity of $\check{\lambda}^t$. Suppose PCI selects a molecule as $\boldsymbol{x}^{t+1}$ from $\boldsymbol{x}^t$'s neighborhood set $\mathcal{N}(\boldsymbol{x}^t)$, where the lowest $\check{\lambda}^{t+1}$ is precisely determined, i.e., finding a solution that maximizes $|\check{\lambda}^{t+1} - \check{\lambda}^t|$.

## A.2 THEOREM

**Theorem A.8** (Approximation Guarantee). *Let the maximum preference-conditioned objective at the t-th iteration $\check{\lambda}^t := \max \{\mathcal{L}_j^t \lambda_j \mid j \in [m]\}$ gain (i.e., $\triangle \check{\lambda}^t$) satisfy submodularity with a curvature ratio $\alpha \in [0, 1]$. Suppose the sizes (i.e., number of substructures) of all the scaffolding trees generated are bounded by $N$. Given an initial molecule $\boldsymbol{x}^0$ and preference vector $\lambda$, PCI algorithm enjoys the following approximation guarantee when performing $T$ optimization rounds:*

$$\mathcal{L}^T \in \mathcal{A} := \left\{ \mathcal{L} \in \mathcal{O} \mid \mathcal{L} \preceq (\Gamma \check{\lambda}^* + (1 - \Gamma)\check{\lambda}^0) \cdot \lambda^{-1} \right\}, \tag{19}$$

*where $\Gamma = \frac{1-\alpha^T}{(1-\alpha)N}$, $\check{\lambda}^* = \max \{\mathcal{L}_j^* \lambda_j \mid j \in [m]\}$, and $\check{\lambda}^0 = \max \{\mathcal{L}_j^0 \lambda_j \mid j \in [m]\}$, and $\lambda^{-1} := (1/\lambda_i, \dots, 1/\lambda_m)$.*

*Proof.* In the following steps of the proof, to simplify mathematical notation, we substitute $\lambda^t$ for $\check{\lambda}^t$. Starting from $\boldsymbol{x}^0$, suppose the path to optimum $\boldsymbol{x}^*$ with the preference $\lambda$ is

$$\boldsymbol{x}^0 \to \boldsymbol{x}^1 \to \boldsymbol{x}^2 \to \cdots \to \boldsymbol{x}^k = \boldsymbol{x}^*, \tag{20}$$

where each step, one substructure is added.

For PCI, we run $T \in [k, N]$ iterations, and the path produced by PCI is

$$\hat{\boldsymbol{x}}^0(\boldsymbol{x}^0) \to \hat{\boldsymbol{x}}^1 \to \hat{\boldsymbol{x}}^2 \to \cdots \to \hat{\boldsymbol{x}}^T, \text{ where } T \geq k. \tag{21}$$

For the optimum $\boldsymbol{x}^*$, based on the submodularity in Assumption A.2 we have

$$k \left(\lambda^0 - \lambda^1\right) \geq \sum_{j=1}^k (\lambda^{j-1} - \lambda^j) = \lambda^0 - \lambda^k = \lambda^0 - \lambda^*. \tag{22}$$

From assumption A.1, it follows that

$$\lambda^0 - \lambda^1 \geq \frac{1}{k}(\lambda^0 - \lambda^*) \geq \frac{1}{N}(\lambda^0 - \lambda^*). \tag{23}$$

For the molecular $\boldsymbol{z}^T$ found by PCI, based on curvature ratio in Assumption A.2 we have

$$\hat{\lambda}^{T-1} - \hat{\lambda}^T \geq \alpha \left(\hat{\lambda}^{T-2} - \hat{\lambda}^{T-1}\right) \geq \cdots \geq \alpha^{T-1} \left(\hat{\lambda}^0 - \hat{\lambda}^1\right). \tag{24}$$

Then we have

$$\hat{\lambda}^0 - \hat{\lambda}^T = \sum_{j=1}^T (\lambda^{j-1} - \lambda^j) \geq \sum_{j=1}^T \alpha^{j-1}(\hat{\lambda}^0 - \hat{\lambda}^1) = \frac{1-\alpha^T}{1-\alpha} \left((\hat{\lambda}^0 - \hat{\lambda}^1)\right). \tag{25}$$

Since PCI pick up a molecule as $\boldsymbol{x}^{t+1}$ from $\boldsymbol{x}^t$'s neighborhood set $\mathcal{N}(\boldsymbol{x}^t)$ with lowest $\check{\lambda}^{t+1}$ is exactly solved, i.e. $\hat{\lambda}^1 \leq \lambda^1$, and $\hat{\lambda}^0 = \lambda^0$. Thus we have

$$\hat{\lambda}^0 - \hat{\lambda}^1 \geq \lambda^0 - \lambda^1. \tag{26}$$

From Eq. 23, Eq. 25 and Eq. 26, it follows that

$$\lambda^0 - \hat{\lambda}^T \geq \frac{1 - \alpha^T}{(1 - \alpha)N}(\lambda^0 - \lambda^*). \tag{27}$$

Thus we have:

$$\hat{\lambda}^T \leq \frac{1 - \alpha^T}{(1 - \alpha)N}\lambda^* + (1 - \frac{1 - \alpha^T}{(1 - \alpha)N})\lambda^0. \tag{28}$$

Finally, it follows that

$$\mathcal{L}^T \in \mathcal{A} := \left\{ \mathcal{L} \in \mathcal{O} \mid \mathcal{L} \preceq (\Gamma\lambda^* + (1 - \Gamma)\lambda^0) \cdot \lambda^{-1} \right\}, \tag{29}$$

$\square$

## B    TIME COMPLEXITY

We did computational analysis in terms of oracle calls and computational complexity.(1) Oracle Calls. PCI has the comparable oracle calls to that of DST, it requires $O(TM)$ oracle calls, where $T$ is the number of iterations (Alg 1). $M$ is the number of generated molecules, we have $M \leq KJ$, $K$ is the number of nodes in the scaffolding tree, for small molecule, K is very small. $J$ is the number of enumerated candidates in each node. (2) Computational Complexity. The computation of $C = G^T G$ has runtime $O(m^2 n)$, where $n$ is the dimension of the gradients. With the current best LP solver [1], our LP (10), that has $m$ variables and at most $2m + 1$ constraints, has a runtime of $O^*(m^{2.38})$. Since in deep networks, usually $n \gg m$, PCI does not significantly increase the computational cost of backpropagation gradient calculation. The complexity and runtime are acceptable for molecule optimization. Furthermore, we present the change in wall clock time when Pareto optimization is incorporated to add one substructure to a initial molecule: $28s \rightarrow 63s$. The introduction of Pareto optimization doesn't increase oracle calls. As the length of molecules becomes longer and longer, the time occupied by oracle calls will dominate.

## C    IMPLEMENTATIONS DETAILS

### C.1    MOLECULAR REPRESENTATION

A molecular graph is a representation of a molecule, consisting of atoms as nodes and chemical bonds as edges. Nonetheless, challenges such as chemical validity constraints, ring integrity, and extensive calculations hinder the explicit reconstruction of potential connectivity. To tackle this, Jin et al. (2018) introduced a scaffolding tree, a spanning tree that employs nodes as *substructures* to model a higher-level representation of a molecule.

A scaffolding tree, denoted by $\mathcal{T}_{\boldsymbol{x}} = \{\mathbf{N}, \mathbf{A}, \mathbf{w}\}$, serves as a high-level representation of a molecule $\boldsymbol{x} \in \mathcal{X}$. Each node is a member of the substructure set $\mathcal{S}$ (also referred to as the vocabulary set). $\mathcal{T}_{\boldsymbol{x}}$ consists of three components: (i) the node indicator matrix defined as $\mathbf{N} \in \{0, 1\}^{K \times |S|}$, where each row of $N$ is a one-hot vector indicating the substructure to which the node belongs; (ii) the adjacency matrix denoted by $\mathbf{A} \in \{0, 1\}^{K \times K}$, where $\mathbf{A}_{ij} = 1$ when the $i$-th and the $j$-th nodes are connected, and 0 when they are unconnected; (iii) $\mathbf{w} = [1, \dots, 1]^{\mathsf{T}} \in \mathbb{R}^K$, signifies that the $K$ nodes are equally weighted. We convert molecules to scaffolding tree for training the surrogate oracle.

To modify the scaffolding tree $\mathcal{T}_{\boldsymbol{x}}$, we employ its differentiable version, $\widetilde{\mathcal{T}}_{\boldsymbol{x}}$, as proposed by Fu et al. Fu et al. (2022). The basic scaffolding tree, $\mathcal{T}_{\boldsymbol{x}}$, can be transformed into a tree containing $K + K_{expand}$ nodes, denoted by $\widetilde{\mathcal{T}}_{\boldsymbol{x}} = \{\widetilde{\mathbf{N}}, \widetilde{\mathbf{A}}, \widetilde{\mathbf{w}}\}$, through the addition of an expansion node set

$\mathcal{V}_{expand} = \{u_v \mid v \in \mathcal{V}_{\mathcal{T}_x}\}$, where $|\mathcal{V}_{expand}| = K_{expand} = K$. It is crucial to note that $\widetilde{\mathcal{T}_x}$ contains learnable parameters, which can be interpreted as conditional probability. This conditional probability can be utilized to sample a new tree through processes such as node shrinking, replacement, or expansion. Each scaffolding tree corresponds to multiple molecules, as substructures can be combined in various ways. We assemble all possible molecules according to Jin et al. (2018).

## C.2 FRAMEWORK DETAILS

We construct a differentiable surrogate model with a Graph Convolutional Network (GCN) (Kipf & Welling, 2016) to capture knowledge from any oracle function. In contrast to DST, which calculates the mean value of multiple properties on the observed dataset $\mathcal{D}$ to derive a single unified property $\widehat{y}$ for training, we individually predict each property score of the molecule. We initialize the representation by $\mathbf{H}^{(0)} = \mathbf{NE} \in \mathbb{R}^{K \times d}$, where $\mathbf{E} \in \mathbb{R}^{|\mathcal{S}| \times d}$ is the embedding matrix of substructures, and is randomly initialized. The updating rule of GCN for the $l$-th layer is:

$$\mathbf{H}^{(l)} = \text{RELU}\left(\mathbf{B}^{(l)} + \mathbf{A}\left(\mathbf{H}^{(l-1)}\mathbf{U}^{(l)}\right)\right), \quad l = 1, \cdots, L, \tag{30}$$

where $L$ is GCN's depth, and $\mathbf{B}^{(l)} \in \mathbb{R}^{K \times d}/\mathbf{U}^{(l)} \in \mathbb{R}^{d \times d}$ are bias/weight parameters. We leverage the weighted average as the readout function of the last layer's node embeddings, followed by multi-head MLP to yield the prediction of $m$ properties:

$$\widehat{y} = \text{MLP}\left(\frac{1}{\sum_{k=1}^{K} w_k} \sum_{k=1}^{K} w_k H_k^{(L)}\right). \tag{31}$$

## C.3 DETAILS OF I-LS

The baseline method Inversion with Linear Scalarization (I-LS) is summaried in Algorithm 2. I-LS dose not compute the non-dominated gradient $d_{nd}$ but instead linearly weights the gradient using preferences, i.e., $d = G * \lambda$. We keep the remaining parts consistent with PCI.

---

**Algorithm 2:** Inversion with Linear Scalarization (I-LS)

---

**Input:** Input molecule $x^0 \in \mathcal{X}$, preference vector $\lambda \in \mathbb{R}^m$, and step size $\eta > 0$.
**Output:** Generated Molecule $x^*$.

1 Initialization.
2 Train surrogate oracle according to Eq. 7.
3 **for** $t = 0, \ldots, T$ **do**
4     Convert molecule $x^t$ to differentiable scaffolding tree $\widetilde{\mathcal{T}}_{x^t}^0$;
5     **for** $k = 0, \ldots, K$ **do**
6         Compute gradients of each property objective w.r.t. $\widetilde{\mathcal{T}}_{x^t}^k$: $G = \nabla\mathcal{L} = [g_1, \ldots, g_m]$;
7         Calculate the direction of descent $d_{ls} = G\lambda$;
8         Update the differentiable scaffolding tree using $\widetilde{\mathcal{T}}_{x^t}^{k+1} = \widetilde{\mathcal{T}}_{x^t}^k - \eta d_{ls}$.
9     **end**
10     Sample discrete $\mathcal{T}_{x^{t+1}}$ from continuous $\widetilde{\mathcal{T}}_{x^t}^K$ and assemble it to molecule $x^{t+1}$.
11 **end**

---

## C.4 PREFERENCE VECTOR

To generate a preference vector uniformly distributed in the objective space, we describe the detailed process. If we want to uniformly sample a point on the arc of unit circle or unit sphere surface, we can follow the steps below:

---

**Algorithm 3:** Generate Preference for Two Objective

---

1    Generate a uniformly distributed variable, $u$, ranging from 0 to 1.
2    Compute coordinates' angle: $\theta = \frac{\pi}{2}u$.
3    Compute Cartesian coordinates: $\lambda_1 = \cos\theta$, $\lambda_2 = \sin\theta$.

---

---

**Algorithm 4:** Generate Preference for Three Objective

---

1    Generate two uniformly distributed variables, $u$ and $v$, ranging from 0 to 1.
2    Compute spherical coordinates' inclination angle and azimuth angle: $\theta = \frac{\pi}{2}u$, $\phi = \arccos v$.
3    Compute Cartesian coordinates: $\lambda_1 = \sin\phi\cos\theta$, $\lambda_2 = \sin\phi\sin\theta$, $\lambda_3 = \cos\phi$.

---

### C.5   PCI SETUP

Most of the settings follow the DST (Fu et al., 2022). We implemented PCI using Pytorch 1.7.1, Python 3.7.9 on an Intel Xeon Platinum 8255C @ 2.50GHz CPU. Both the size of substructure embedding and hidden size of GCN (GNN) are $d = 100$. The depth of GNN $L$ is 3. In each generation, we keep $C = 10$ molecules for the next iteration. The learning rate is 1e-3 in training and inference procedure. we set the iteration $T$ to a large enough number and tracked the result. When oracle calls budget is used up, we stop it. All results in the tables are from experiments up to $T = 50$ iterations. For ILS, We only replace the objective function in PCI with Linear Scalarization, and other settings are consistent with PCI.

### C.6   ASSEMBLE THE SCAFFOLDING TREE INTO MOLECULE

(a) Ring-atom connection. When connecting atom and ring in a molecule, an atom can be connected to any possible atoms in the ring. Ring-ring connection. (b) When connecting ring and ring, there are two general ways, (1) one is to use a bond (single, double, or triple) to connect the atoms in the two rings. (2) another is two rings share two atoms and one bond.

### C.7   BASELINES

In this section, we describe the experimental setting for baseline methods. Most of the settings follow the original papers.

- **LigGPT** is a string-based distribution learning model with a Transformer as decoder (Bagal et al., 2021), we trained it for 10 epochs using the Adam optimizer with a learning rate of $6e - 4$.

- **GCPN** (Graph Convolutional Policy Network) (You et al., 2018) leveraged graph convolutional network and policy gradient to optimize the reward function that incorporates target molecular properties and adversarial loss. we trained it using Adam optimizer with 1e-3 initial learning rate, and batch size is 32.

- **MolDQN** (Molecule Deep Q-Networks) (Zhou et al., 2019) formulate the molecule generation procedure as a Markov Decision Process (MDP) and use Deep Q-Network to solve it. Adam is trained Adam optimizer with 1e-4 as the initial learning rate, $\epsilon$ is annealed from 1 to 0.01 in a piecewise linear way.

- **GA+D** (Genetic Algorithm with Discriminator network) (Nigam et al., 2020) uses a deep neural network as a discriminator to enhance exploration in a genetic algorithm and is trained using the Adam optimizer with a learning rate of $1e - 3$, $\beta$ is set it to 10.

- **MARS** (Xie et al., 2021) leverage Markov chain Monte Carlo sampling (MCMC) on molecules with an annealing scheme and an adaptive proposal. It is trained using Adam optimizer with 3e-4 initial learning rate.

- **RationaleRL** (Jin et al., 2020) is a deep generative model that grows a molecule atom-byatom from an initial rationale (subgraph). It is trained using Adam optimizer on both pre-training and fine-tuning with initial learning rates of 1e-3, 5e-4, respectively. The annealing rate is 0.9.

- **ChemBO** (chemical Bayesian optimization) (Korovina et al., 2020) leverage Bayesian optimization. It also explores the synthesis graph in a sample-efficient way and produces synthesizable candidates. Following the default setting in the original paper, the number of steps of acquisition optimization is set to 20. The initial pool size is set to 20, while the maximal pool size is 1000.

- **BOSS** (Bayesian Optimization over String Space) (Moss et al., 2020) builds a Gaussian process surrogate model based on Sub-sequence String Kernel, which naturally supports SMILES strings with variable length, and maximizing acquisition function efficiently for spaces with syntactical constraints. The population size is set to 100, the generation (evolution) number is set to 100.

- **DST** (Differentiable Scaffolding Tree) (Fu et al., 2022) utilizes a learned knowledge network to convert discrete chemical structures to locally differentiable ones. DST enables a gradient-based optimization on a chemical graph structure by back-propagating.

- **MOGFN-PC** (Preference-conditional GFlowNets) (Jain et al., 2023) is a Reward-conditional GFlowNets based on Linear Scalarization. They introduce the Weighted-log-sum that can help in scenarios where all objectives are to be optimized simultaneously, and the scalar reward from Weighted-Sum can be dominated by a single reward.

- **RetMol** (Retrieval-Based Molecular Generation) (Wang et al., 2023) retrieves and fuses the exemplar molecules with the input molecule, which is trained by a new selfsupervised objective that predicts the nearest neighbor of the input molecule.

## C.8 N-GRAMS

The task is to generate sequences of some maximum length $L$, which we set to 36 for the experiments. We consider a vocabulary (actions) of size 21, with 20 characters ["A", "R", "N", "D", "C", "E", "Q", "G", "H", "I", "L", "K", "M", "F", "P", "S", "T", "W", "Y", "V"] and a special token to indicate the end of sequence. The rewards $\{R_i\}_{i=1}^d$ are defined by the number of occurrences of a given set of n-grams in a sequence x. For instance, consider ["AB", "BA"] as the n-grams. The rewards for a sequence x = ABABC would be [2, 1]. In our experiments, we use 3 Unigrams ["A", "C", "V"] and 3 Bigrams ["AC", "CV", "VA"].

## D ADDITIONAL EXPERIMENTAL RESULTS

### D.1 MOLECULES GENERATED BY PCI

We provide several molecules synthesized via the PCI approach.

(1) **Molecules with JNK3 and GSK3$\beta$ scores**. Each score independently represents the respective values for JNK3 and GSK3$\beta$, see Figure. 7.

(2) **Molecules with highest average QED, normalized-SA, JNK3 and GSK3$\beta$ scores**. These four scores symbolize the values for QED, normalized SA, JNK3, and GSK3$\beta$, respectively, see Figure. 8.

### D.2 MOLECULES GENERATED BY LINEAR SCALARIZATION

We exhibit the molecular graphs produced through Linear Scalarization. Alongside these visual representations, their corresponding property scores are included, and the loss ratio is calculated using the formula $ratio = \frac{l_{JNK3}}{l_{GSK3\beta}}$. This supplementary information further elaborates on the experimental results outlined in Section 6.3 of the paper, as illustrated in Figure. 9.

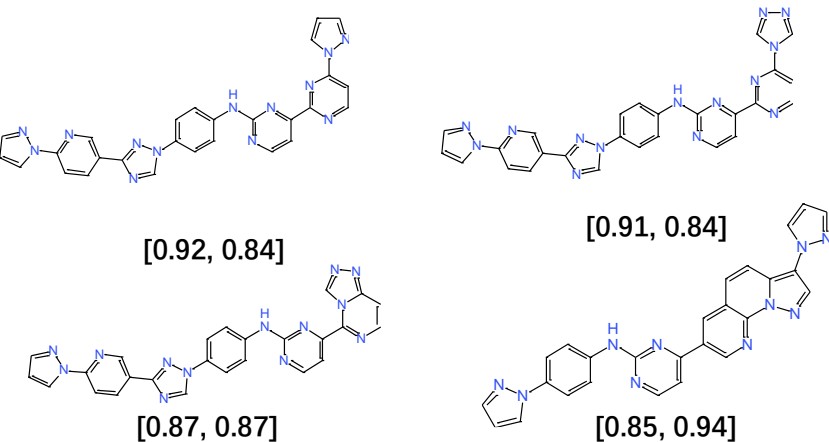

[0.92, 0.84]

[0.91, 0.84]

[0.87, 0.87]

[0.85, 0.94]

Figure 7: Generated molecules by PCI. These four scores symbolize the values for JNK3 and GSK3$\beta$, respectively.

**[0.76, 1, 0.7, 0.87]**

**[0.75, 1, 0.74, 0.84]**

**[0.75, 1, 0.69, 0.87]**

**[0.77, 1, 0.68, 0.85]**

**[0.77, 1, 0.71, 0.82]**

**[0.71, 1, 0.72, 0.87]**

Figure 8: Generated molecules by PCI. These four scores symbolize the values for QED, normalized SA, JNK3, and GSK3$\beta$, respectively.

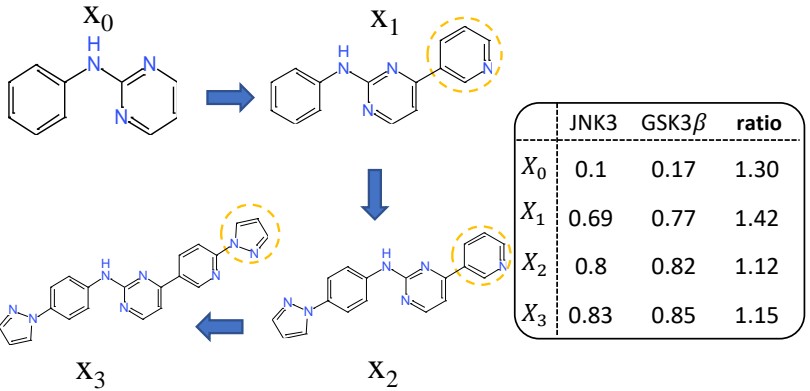

| | JNK3 | GSK3$\beta$ | ratio |
|---|---|---|---|
| $X_0$ | 0.1 | 0.17 | 1.30 |
| $X_1$ | 0.69 | 0.77 | 1.42 |
| $X_2$ | 0.8 | 0.82 | 1.12 |
| $X_3$ | 0.83 | 0.85 | 1.15 |

Figure 9: Generated molecules by I-LS, property scores and loss ratio.

