# OpenReview forum: "Trading-off Multiple Properties for Molecular Optimization"
_ICLR.cc/2024/Conference — Submitted to ICLR 2024_

### Official Review · Reviewer_8aLW · 2023-10-20

**Soundness:** 2 fair
**Presentation:** 3 good
**Contribution:** 2 fair
**Rating:** 3
**Confidence:** 4

**Summary:**

The paper introduces a multi-objective optimization technique for molecular structures, termed preference-conditioned inversion (PCI). Experimentally, PCI outperforms traditional linear scalarization in probing the Pareto frontier. Nevertheless, its central concept isn't groundbreaking, having been previously documented in [1]. The non-trivial effort of converting the molecular design challenge into gradient-based optimization is attributed to [2]. While the work has merit, its technical novelty seems modest. The manuscript might be better suited for a chemistry-focused journal.

References:
[1] Platt, J., & Barr, A. (1987). Constrained differential optimization. In Neural Information Processing Systems.
[2] Fu, T., Gao, W., Xiao, C., Yasonik, J., Coley, C. W., & Sun, J. (2021). Differentiable scaffolding tree for molecular optimization. arXiv preprint arXiv:2109.10469.

**Strengths:**

## Quality of the writing
The method is articulated lucidly.

## Significance of the problem
The task of navigating the Pareto frontier in molecular optimization is pivotal for various subsequent applications.

**Weaknesses:**

## Question of Novelty
The manuscript's primary shortcoming lies in its novelty. As mentioned earlier, the foundational concept was presented in 1987 [1], and an enabling step for translating the molecular design task to gradient-based optimization was introduced in [2]. Despite references to many SOMO methods favoring linear scalarization for multi-objectivity, I think the reason they chose linear scalization is that the their emphasis is on the algorithm development while an extension to MOMO is relatively trivial. It might be more apt to redirect this focus towards application.

## Concerns over Linear Scalarization Critique
The authors critique linear scalarization methods, arguing that they can't comprehensively traverse the Pareto solution merely by weight adjustments, as depicted in Figure 3(b). Contrarily, based on my experience, linear scalarization can navigate along the Pareto front toward both extremes, and with weight tuning and keeping all points visited during the processes, results comparable to PCI's can be achieved. It would be beneficial for the authors to record all points traversed during optimization and employ hypervolume as a performance metric.

## Generalizability of the Method
DST represents only a handful of cases that render the molecular design problem differentiable, enabling the application of gradient-based methods. The value of PCI is limited if it can not be adapted for broader molecular design techniques.

## Lack of thorough investigation
The authors recurrently state that MOMO's aim is to scrutinize the Pareto front, yet they predominantly highlight a single value representing an average of the top-100 molecules. A comparison of the Pareto fronts analyzed by PCI and LS would bolster their claims.

## No open-sourced code is available

**Questions:**

- In Figure 1(b), are the trajectories experimentally derived or merely illustrative?
- In Table 2, what is the rationale behind allocating 5 oracle calls for Graph GA to train surrogates?

---

> ### Author Response · Authors · 2023-11-14
>
> Thank you for taking the time to review our paper and providing valuable feedback. We would like to answer your questions below.
>
> **1. [W1] The multi-objective optimization problem is not trivial.**
>
> **Firstly, we believe it is not appropriate to dismiss the research field of multi-objective optimization as trivial.** General multi-objective optimization is currently an important research direction in machine learning, with numerous efforts dedicated to solving Pareto optimization problems. And Linear Scalarization is considered a weak baseline in multi-objective optimization. Moreover, the Pareto optimization problem remains unresolved in molecular optimization.
>
> **Secondly, from SOMO to MOMO is the research trend in drug design.** Previous works usually focused on single properties. However, it is until this year that we have witnessed the emergence of tasks involving multi-objective molecular optimization in machine learning conferences (ICML 2023 [1] and NeurIPS 2023 [2]). Futhermore, they adopt Linear Scalarization or Bayesian optimization to tackel this problem, which is the baseline of our method.
>
> **Thirdly, Thirdly, thanks for reviewer xqpc and reviewer dF44 to recognize the contribution of our work.** Reviewer xqpc finds our work novel and interesting, and Reviewer dF44 considers it intriguing and a practically valuable contribution.
>
> **2. [W2 & Q1] About Linear Scalarization Critique**
>
> It has been mathematically proved by [3]  in Chapter 4.7.4 and explaining why linear scalarization fails to capture preference-conditioned solutions.	Our experiments also confirm this phenomenon. We provide the code for you to reproduce our Figure 1 and Figure 3 within few minutes. To avoid any other factors, we did not employ any tricks or set random seeds. The drawbacks of Linear Scalarization are quite evident as we mentioned in our paper. Please refer to the provided code for further details: https://github.com/ICLR2024/PCI
>
> **Our code can also compute the hypervolume:**
>
> **PCI: 0.33        I-LS: 0.065**
>
> You stated ‘’Contrarily, based on my experience, linear scalarization can navigate along the Pareto front toward both extremes, and with weight tuning and keeping all points visited during the processes, results comparable to PCI's can be achieved.‘’ Could you please provide some evidence to support your viewpoint? Or could you please point out how to improve our code to achieve 'Linear scalarization's results are comparable to PCI'.
>
> **3. [W3] Generalizability of the Method.**
>
> If the problem is non-differentiable, we can train a differentiable surrogate model and apply PCI.
>
> **4. [W4] About Top-K molecules.**
>
> Top-k is a commonly used metric in molecular optimization, and we are simply following the existing works[1][4][5], as we have already mentioned in our paper. Generally, in molecular optimization, people usually evaluate the top points, which is more normal in optimization (since one generally cares about the optimum reached, not how one gets there). For instance, if we need to add 5 substructures to achieve a high property score, why do we need to consider the molecules (3 substructures) generated during the intermediate steps as part of the final result and include them in the statistics? In addition, we set K to 100 instead of the commonly used values of 3 or 10 in existing works, which makes our result more convincing.
>
> **5. [Q2] Oracle calls for Graph GA.**
>
> We apology for the typo. We have already corrected it to 0+25k.
>
> **Reference:**
>
> [1] Moksh Jain, Sharath Chandra Raparthy, Alex Hernandez-Garcıa, Jarrid Rector-Brooks, Yoshua Bengio, Santiago Miret, and Emmanuel Bengio. Multi-objective gflownets. In International Conference on Machine Learning, pp. 14631–14653. PMLR, 2023.
>
> [2] Zhu Y, Wu J, Hu C, et al. Sample-efficient Multi-objective Molecular Optimization with GFlowNets[J]. arXiv preprint arXiv:2302.04040, 2023.
>
> [3] Boyd, S. and Vandenberghe, L. Vector optimization. In Convex Optimization, chapter 4.7, pp. 174–187. Cambridge University Press, 2004.
>
> [4] Emmanuel Bengio, Moksh Jain, Maksym Korablyov, Doina Precup, and Yoshua Bengio. Flow network based generative models for non-iterative diverse candidate generation. Advances in Neural Information Processing Systems, 34:27381–27394, 2021.
>
> [5] Tianfan Fu, Wenhao Gao, Cao Xiao, Jacob Yasonik, Connor W Coley, and Jimeng Sun. Differentiable scaffolding tree for molecule optimization. In International Conference on Learning Representations, 2022.

---

> > ### Comment · Reviewer_8aLW · 2023-11-20
> > **The authors seem to have misunderstood my review**
> >
> > Firstly, it's important not to conflate concepts. My commentary specifically addresses this work, not the broader field of multi-objective optimization. I don’t think this paper is proposing a novel multi-objective optimization method, even restricted to the field of molecular optimization, which is my main concern here.
> >
> > The primary concern lies in the unclear articulation of the work's relationship to previous studies, the novelty of your research, and its suitability for this venue. Overall, this work appears to be based on the DST SOMO method, combined with an established multi-objective approach. **Neither element of the MOMO—extension to multi-objective and backbone molecular optimization method—is an original contribution of this paper.** However, the connection with previous methods is not well-represented in the paper. Similar works are typically published in chemical journals, not AI conferences. Regarding the two GFlownet MOMO papers the authors mentioned, it’s essential to recognize that even published works are not flawless. Those papers exhibit similar issues, and I would raise the same concerns if I were reviewing them. Therefore, their acceptance doesn't justify the acceptance of your work. Given the limited technical improvements and lack of significant effort in domain adaptation, I don’t believe this paper is suitable for publication in ICLR.
> >
> > Efficiently exploring the Pareto front in MOMO methods is undoubtedly valuable. However, the appropriate comparison should be with linear scalarization using SOMO, scanning different weights, and evaluating the molecules obtained across the entire trajectory to assess the Pareto front, not just considering the point of convergence. I already mentioned this in my original review, but since the author seems to be asking again, I emphasize it once more.
> >
> > Lastly, regarding generality, I am referring to the multi-objective optimization method added on top of DST, which seems to be the focus of this paper. Among all molecular optimization methods, differentiable approaches are few. Therefore, the method discussed in this paper is not readily applicable to other molecular optimization methods. This is a disadvantage compared to more common approaches like linear scalarization or non-dominant sorting. The author's response about "training a differentiable surrogate model" refers to generalization across optimization objectives, which are enabled by DST, not by the work presented in this paper.

---

### Official Review · Reviewer_dF44 · 2023-11-01

**Soundness:** 3 good
**Presentation:** 3 good
**Contribution:** 3 good
**Rating:** 5
**Confidence:** 4

**Summary:**

For molecular optimization in drug discovery, it is essential to simultaneously optimize multiple properties that may either contradict or correlate with each other, such as efficacy, potency, safety, bioavailability, and ease of synthesis. This paper proposes a framework called Preference-Conditioned Inversion (PCI) for multi-objective molecular optimization in such discrete spaces, given a set of m objective properties with specified preferences. Firstly, a surrogate function that provides m objective function values for a given molecular graph is learned using the differential scaffolding tree (Fu et al, 2022). Then, within the convex hull of the gradient vectors of the m objective characteristics, the direction that most aligns with the given preference vector and moves towards the Pareto optimal solution is determined by solving a linear programming problem updating the differential scaffolding tree. After several iterations of this update, the discrete scaffolding tree is sampled and assembled into molecules, updating molecule x. By repeatedly updating in this manner, a Pareto optimal solution that aligns with the preference in the discrete molecular graph space is generated. When enumerating Pareto optimal solutions, they can be efficiently generated by systematically changing the preference. In the paper, the effectiveness of this method is demonstrated by first performing a sanity check on a synthetic task and then comprehensively comparing it with many baselines in an actual molecular optimization task.

**Strengths:**

In optimization with multiple objective functions, it is standard to use Linear Scalarization, which rewrites them into a single objective function by weighted sum based on a preference vector, and then optimize it. In discrete optimization like molecular optimization, which is the subject of this paper, it is known that the solutions generated by this Linear Scalarization fail to capture trade-offs and produce biased solutions. This point is also illustrated in Figure 1. On the other hand, evolutionary computation and multi-objective Bayesian optimization have issues in terms of computational efficiency and alignment with the given preference vector. The proposed method addresses the demonstrated shortcomings of these existing methods, providing an algorithm that efficiently searches for Pareto optimal solutions that align well with the given preferences, making it a practically valuable contribution.

Firstly, the paper learns m objective characteristics using differentiable methods and then freezes them. The idea of leveraging their differentiability to compute multiple gradient vectors and explicitly optimizing their convex combination with linear programming, and then sampling the locally optimal solutions to return to discrete optimization is technically very intriguing. The experimental results, which include comparisons with many methods including Linear scalarization on actual multi-objective molecular optimization benchmarks, show good performance.

**Weaknesses:**

The proposed method deals with molecular optimization given a preference vector, and a comparison with Linear scalarization is of primary interest. However, on several points, it is unclear why the proposed method is superior to Linear scalarization:

1. The procedure for the non-dominating descent direction by linear programming, which is the core of the proposed algorithm, Eq.(9), is designed based on the non-uniformity criterion, eq.(3). But this metric would not be assumed in Linear Scalarization. While it seems natural that the score for eq.(3) is better than LS if we assumed this scoring, the practical significance of this is unclear.

2. Although this method is proposed as a framework, it heavily relies on the "differentiable scaffolding tree" method (Fu et al, 2022). It's unclear if this specific method is essential, or if other surrogate methods could also be suitable. Moreover, especially the procedure in the inversion step, "sample the discrete scaffolding tree", is not clearly described. A more detailed comparison with the experimental results of I-LS and PCI would be appreciated for readers to interpret the comparisons between I-LS and PCI.

3. This study seems to specialize insights from established multi-objective optimization research, specifically De ́side ́ri et al (2012) and Mahapatra & Rajan (2020), to multi-objective optimization in the discrete space of molecular optimization. If we assume that this work utilizes the existing "differential scaffolding tree" (Fu et al, 2022) and its sampling function as a differentiable surrogate for this purpose, its technical contribution would appears somewhat incremental.

**Questions:**

Q1. If the alignment to the given reference is evaluated using methods other than eq.(3), can we say that PCI is better than I-LS? Is this criteria of eq.(3) appropriate for this comparison?

Q2. Given that the paper's focus is on molecular optimization with a given preference, it seems natural to have multiple metrics. Are there no metrics other than the non-uniformity in eq.(3)?

Q3. What is the procedure for the "sample the discrete scaffolding tree" part? This process involves discretization, but can its effects be ignored?

Q4. This study heavily relies on the existing method of the differentiable scaffolding tree (Fu et al, 2022). Is it essential to use this method, or can the surrogate part be replaced with other methods? If it can be replaced, why was only the differentiable scaffolding tree used?

Q5. Is the standpoint of this study to specialize insights from general multi-objective optimization, namely De ́side ́ri et al (2012) and Mahapatra & Rajan (2020), for multi-objective optimization in the discrete space of molecular optimization? Or does it also provide new insights into general multi-objective optimization?

---

> ### Author Response · Authors · 2023-11-14
>
> Thank you for taking the time to review our paper and providing valuable feedback. We would like to answer your questions below.
>
> **1. [Q1 & Q2] About Non-Uniformity.**
>
> Before we discuss the Non-Uniformity, I would like to share an example I provided to Reviewer xqpc, which effectively demonstrates the importance of Non-Uniformity in MOMO and the advantage of PCI.
>
> For instance, let's consider two properties (A, B) with a solution space of {(1, 1), (1, 0.5), (1,0.33), (0, 0)}. We aim to optimize these attributes using SOMO+scalarization, starting from the initial point (0,0). **Assuming that a very high value of property B has a negative impact, we need to maintain it at a moderate level.** Thus we assign a preference ratio of 2:1 and the ideal optimal solution is (1,0.5), and define the unified attribute as P = A + 0.5B. However, for the solution (1,1), P = 1 + 0.5 = 1.5. For the solution (1,0.5), P = 1 + 0.5 * 0.5 = 1.25. Clearly, 1.5 > 1.25, and we end up with the solution (1,1) instead of the desired (1,0.5). Even if we change the weight to 3:1, we still obtain the same solution (1,1) instead of desired (1,0.33). However, PCI can find the desired solution (1, 0.5) and (1,0.33) since we take preference condition and decent direction in to consideration by solving Eq. 9. We have demonstrated this through both theoretical analysis and experimental validation. **Due to the limitations of the response's length, we strongly recommend reviewer to refer to a blog [1] posted by DeepMind in 2021 which illustrates why linear scalarization often results in biased solutions and fails to adjust scalar weights in an easy-to-understand manner.**
>
> **1.1 [Q2] Why is Non-Uniformity?**
>
> Non-uniformity is a metric used to measure the similarity between the ratio of properties and  preference. Therefore, any metric used to measure the similarity between two vectors can be employed, such as cosine similarity. We use non-uniformity because we follow the general multi-objective optimization.
>
> **1.2 [Q1] Non-Uniformity is appropriate for MOMO.**
>
> In the aforementioned example, if the preference is 2:1, scalarization fails to find the desired solution (1,0.5) , while PCI can. It is evident that (1,0.5) outperforms (1,1) in terms of Non-Uniformity. **In practical drug design, it is not always desirable for certain properties to be maximized or minimized. Instead, we may need to control them within a certain range with preference. However, scalarization fails to achieve this level of control.** Such requirements are quite common in drug design and remain unresolved.
>
> **2. [Q3] About Discretization.**
>
> Our sampling process is consistent with the differentiable scaffolding tree  (DST), as detailed in [1]. After discretization, like other molecular optimization methods, we employ oracle to test the discretized molecules and select those that meet the conditions (maximum relative objective value $\check{\lambda}^t$ decreased) while discarding the ones that do not. As a result, the final set of molecules will also satisfy the conditions.  **It is worth noting that in our theoretical analysis, we directly analyze the discretized molecules instead of the continuous form.**
>
> **3. [Q4] Can the surrogate part (DST) be replaced with other methods?**
>
> **The surrogate part can be replaced, but the surrogate part should satisfy some conditions.**  PCI requires the surrogate part to be differentiable, which is why current Combinatorial Optimization (CO) algorithms are not suitable. Another category, Constrained Generative Models (CGMs), conducts optimization in a continuous latent space that is differentiable. However, achieving an ideal smooth and discriminative latent space, typically required by CGMs, has proven to be challenging in practice and CGMs do not perform well in molecular optimization [3][4], which is also shown in our Table. 2. **Therefore, currently the most suitable surrogate part is DST, as it allows us to directly modify the molecular structure and is differentiable.** The current backbones are all developed based on SOMO, but developing a backbone that is more suitable for MOMO could be an interesting future work.

---

> ### Author Response · Authors · 2023-11-14
>
> **4. [Q5] Is the standpoint of this study to specialize insights from general multi-objective optimization? Or does it also provide new insights into general multi-objective optimization?**
>
> Yes, the disadvantage of linear scalarization was first argued in general multi-objective optimization and this problem is still unsolved in molecular optimization task.  And PCI also provide new insights into general multi-objective optimization. Currently, general multi-objective optimization focuses on continuous spaces, especially the continuous parameters of deep models. It is still unclear how to perform multi-objective optimization in discrete spaces instead of using  black-box search methods such as Genetic Algorithms and Bayesian Optimization. Technically, we have transformed the discrete problem into a locally differentiable problem and provided convergence guarantees. Furthermore, we believe that drawing insights from general multi-objective optimization and addressing an still unsolved and important problem in molecular optimization is both meaningful and novel.
>
> **Reference:**
>
> [1] Jonas Degrave, Ira Korshunova. Why machine learning algorithms are hard to tune. (https://www.engraved.blog/why-machine-learning-algorithms-are-hard-to-tune/)
>
> [2] Tianfan Fu, Wenhao Gao, Cao Xiao, Jacob Yasonik, Connor W Coley, and Jimeng Sun. Differentiable scaffolding tree for molecule optimization. In International Conference on Learning Representations, 2022.
>
> [3] Nathan Brown, Marco Fiscato, Marwin HS Segler, and Alain C Vaucher. Guacamol: benchmarking models for de novo molecular design. Journal of chemical information and modeling, 59(3): 1096–1108, 2019.
>
> [4] Kexin Huang, Tianfan Fu, Wenhao Gao, Yue Zhao, Yusuf Roohani, Jure Leskovec, Connor W Coley, Cao Xiao, Jimeng Sun, and Marinka Zitnik. Therapeutics data commons: Machine learning datasets and tasks for drug discovery and development. arXiv preprint arXiv:2102.09548, 2021.

---

> ### Author Response · Authors · 2023-11-20
>
> Dear Reviewer dF44,
>
> Once again, we appreciate your commitment and valuable feedback during the review of our work. We have provided responses to all your concerns. In summary:
>
> - We explained why non-uniformity is appropriate for comparison in MOMO, and illustrate why PCI is better than I-LS with an example;
> - We provided details about procedure for the "sample the discrete scaffolding tree" part;
> - We also clarified our convergence analysis is based on discrete molecules. We have take the discretization into consideration;
> - We explained that the surrogate part should satisfy some conditions to replace DST;
> - We provided analysis about the relationship between PCI and general multi-objective optimization, and the new insights to conduct Pareto optimization in discrete space.
>
> If there are any more questions, please let us know. We hope that you will take into account our previous responses when making your final assessment of our work!

---

> ### Comment · Reviewer_dF44 · 2023-11-21
>
> Thank you for the detailed feedback! The responses are informative.
>
> Still, I felt that this work is an application of some existing ideas to a very specific situation where "the goal is multi-objective but a preference for the objectives is explicitly given". The answers to Q1-Q4 made me feel that the presented idea doesn't sound like a novel framework, but rather it is just presenting a combined use of several specific ideas in this specific situation, specifically, differentiable scaffolding tree (DST) by Fu et al, 2022 and MOMO ideas by De ́side ́ri et al (2012) and Mahapatra & Rajan (2020).
> Also, the given answers to Q5 was still unconvincing to me. (with respect to the concerns in "Weakness")
>
> **A question**:
>
> I didn't quite catch what you mean by the presented example of "two properties (A, B) with a solution space of {(1, 1), (1, 0.5), (1,0.33), (0, 0)}". So can I have more explanations on this?
>
> Actually, this example was hard to interpret, and just raises many many questions: What do you mean by solution space? What do you assume as the Pareto front in this case? Is the tuple like (1,1) the loss or the input variable? What is the given preference in this case? This is also needed for linear scalarization, right? What does "starting from the initial point (0,0)" mean? What is "initial point" here for linear scalarization in this example? What do you mean by "Assuming that a very high value of property B has a negative impact, we need to maintain it at a moderate level." Why is only this part qualitative? Is this about a loss or the value of objective functions? What do you mean by "PCI can find the desired solution (1, 0.5) and (1,0.33) since we take preference condition"? What is the preference condition in this example? ...

---

### Official Review · Reviewer_xqpc · 2023-11-03

**Soundness:** 3 good
**Presentation:** 3 good
**Contribution:** 3 good
**Rating:** 5
**Confidence:** 4

**Summary:**

This paper proposes PCI (preference-conditional inversion) framework that aims to efficiently find the Pareto optimal solution for multi-objective (i.e., multi-property) molecular design that matches a given preference across the different objectives.
This is achieved by first training a differentiable surrogate model for the oracle, which is then used to guide the molecular optimization process.
In each iteration of this optimization process, PCI computes the non-dominating gradient and performs gradient descent to identify preference-conditioned "local" Pareto optimal solution, which is used to sample a discrete scaffolding tree and assemble it to the molecule.
Based on a synthetic example as well as a multi-objective molecular optimization task with various combinations of properties, the paper shows that the proposed PCI may provide an efficient way of exploring trade-offs between multiple properties in molecular optimization problems.

**Strengths:**

This paper proposes a novel approach, called PCI (preference-conditioned inversion), for multi-objective optimization, which may enable efficient search of Pareto optimal solutions that meet given preference vectors.
The construction of a differentiable surrogate oracle that can be used to efficiently identify local Pareto optimal solutions, which then can be used to iterative assemble optimized molecules that are Pareto optimal and meet the preference conditions is novel and interesting.
By comparing with other popular schemes for multi-objective molecular optimization, the paper motivates the proposed scheme and illustrates its potential benefits.
The synthetic example as well as the molecular optimization design tasks with various combinations of molecular properties (that are frequently used to evaluate molecular design algorithms) demonstrate the potential advantages of the proposed scheme.
The evaluations show that PCI effectively optimizes the property scores of top molecules outperforming all other alternatives considered in this study..
Furthermore, PCI shows good performance in terms of diversity and novelty, outperforming almost all (except for LigGPT in terms of diversity).

**Weaknesses:**

While the paper is overall well-written, there are a number of issues that need to be addressed.

1. It is unclear whether PCI will provide any clear advantage over other SOMO schemes applied to MOMO tasks via linear scalarization - *IF* one has a specific preference condition to impose on the multiple properties to be optimized.
In such a case, would there be any advantage for PCI compared to MOMO method via SOMO+scalarization?

2. Similarly, if one has not a single but still a "small" number of preference conditions the optimized molecules need to meet, what would be the advantage of using PCI compared to repeating MOMO via SOMO+scalarization for each of the preferences?

3. However, when one desires to explore the overall Pareto front for a variety of preferences, it seems that the proposed PCI scheme may begin to provide distinct advantages over MOMO via SOMO+scalarization, since the pre trained surrogate oracle can be put to good use for a large number of different preferences - despite the initial cost (e.g., in terms of oracle calls) of training the surrogate.
It would be very helpful if the authors could discuss when PCI would have clear advantages over other simple extensions of SOMO methods.

4. In Fig. 1(b), it is not very intuitive why linear scalarization would result in the optimization trajectories shown in the figure.
Please provide a detailed explanation of the LS optimization scheme that is illustrated in Fig. 1(b) and provide some insights as to why the scheme would prefer Pareto optimal solutions located near the end of the allowable preference regions.
This also applies to the results shown in Fig. 3(b).

5. I-LS is defined as a scheme that uses "the same inversion framework as PCI but adopts linear scalarization".
This is somewhat ambiguous, and it would be helpful if the authors could refer to the PCI diagram in Fig. 2 and clearly explain which part is changed and how.

6. I-LS seems to perform surprisingly well in the evaluations (e.g., Table 1 and Table 2), where PCI doesn't necessarily outperform I-LS by a significant margin for all performance metrics.
This again raises questions about the fundamental advantages of PCI over simpler scalarization-based approaches (e.g., comments #1-#3 above).

**Questions:**

Please refer to the questions in the Weaknesses section above.

---

> ### Author Response · Authors · 2023-11-14
>
> We thank you for recognizing the novelty and contributions of our paper and also for the positive feedback. We would like to answer your questions below.
>
> **1. Why is Linear Scalarization suboptimal and How does PCI outperform It?**
>
> &emsp;**1.1 [W4] An intuitive understanding of the disadvantage of Linear Scalarization.**
>
> We provide an example to illustrate the limitations of Linear Scalarization, which also serves as the inspiration behind our work. For instance, let's consider two properties (A, B) with a solution space of {(1, 1), (1, 0.5), (1,0.33), (0, 0)}. We aim to optimize these attributes using SOMO+scalarization, starting from the initial point (0,0). **Assuming that a very high value of property B has a negative impact, we need to maintain it at a moderate level.** Thus we assign a preference ratio of 2:1 and the ideal optimal solution is (1,0.5), and define the unified attribute as P = A + 0.5B. However, for the solution (1,1), P = 1 + 0.5 = 1.5. For the solution (1,0.5), P = 1 + 0.5 * 0.5 = 1.25. Clearly, 1.5 > 1.25, and we end up with the solution (1,1) instead of the desired (1,0.5). Even if we change the weight to 3:1, we still obtain the same solution (1,1) instead of the desired (1,0.33).
>
> The example intuitively illustrates why SOMO+scalarization fails to achieve the desired preference-based solutions, as shown in Fig. 1(b). Additionally, even when we change the preference, the obtained solutions remain unchanged and located near the end of the preference regions, as shown in Fig. 3(b). **Such requirements are quite common in drug design and remain unresolved in current scalarization-based molecular optimization methods. Due to the limitations of the response's length, we strongly recommend reviewer to refer to a blog [1] posted by DeepMind in 2021 which  illustrates why linear scalarization often results in biased solutions and fails to adjust scalar weights in an easy-to-understand manner.**
>
> Mathematically, A well-known literature [2] on optimization offers a formal mathematical analysis in Chapter 4.7.4, explaining why linear scalarization fails to capture preference-conditioned solutions.
>
> &emsp;**1.2 [W1 & W2 & W3] In the questions you mentioned, PCI exhibits clear advantages over SOMO + scalarization.**
>
>  &emsp;**(1) [W1] A specific preference:** If the preference is 2:1, SOMO+scalarization often result in biased solutions (1,1). On the other hand, PCI can identify (1,0.5) since we take preference condition and decent direction in to consideration by solving Eq. 9. We have demonstrated this through both theoretical analysis and experimental validation.
>
> &emsp;**(2) [W2] A "small" number of preference:** For example, we use preference 2:1 and 3:1. PCI can obtain corresponding Pareto optimal solutions (1,0.5) and (1,0.33) based on different preferences. However, SOMO+scalarization tends to generate biased solutions, and using different preferences may still result in the same solution (1,1).
>
> &emsp;**(3) [W3] A variety of preferences:** In the example above, despite using a variety of preferences, SOMO+scalarization consistently misses the points (1,0.5) and (1,0.33). This corresponds to Fig. 3(b), where there is still a certain unexplored space on the Pareto front. However, PCI, being capable of generating preference-conditioned solutions, can approximately cover the overall Pareto front.
>
> **2. [W5] The Details About I-LS.**
>  In Fig. 2, I-LS dose not compute the non-dominated gradient $d_{nd}$ but instead linearly weights the gradient using preferences, i.e.,  $d=G*\lambda$. We keep the remaining parts consistent with PCI. We have included the algorithm for I-LS in Section C.3 of the appendix, highlighted in blue.
>
> **3. [W6] About experimental results.**
> To ensure fairness of comparison, both PCI and I-LS employ DST as the backbone network. Since DST itself performs exceptionally well, I-LS also outperforms many baselines. However, we should focus on the relative improvement of PCI and I-LS over DST, which indicates that PCI outperforms I-LS by a significant margin across all performance metrics. The relative improvement is shown in the table below.
>
> &emsp;&emsp;&emsp;GSK3$\beta$+JNK3  &emsp;GSK3$\beta$+JNK3+QED+SA
> | | Nov | Div | APS | Nov | Div | APS |
> | :----: | :----: | :----: | :----: | :----: | :----: | :----: |
> | I-LS | 0% | -5.7% | -0.4% | 0%| -5.1% |-1.8%|
> | PCI | 0%| 1.8% | 1.4% | 0% | 1.4% | 2.1%|
>
>
> **Reference:**
>
> [1] Jonas Degrave, Ira Korshunova. Why machine learning algorithms are hard to tune. (https://www.engraved.blog/why-machine-learning-algorithms-are-hard-to-tune/)
>
> [1] Boyd, S. and Vandenberghe, L. Vector optimization. In Convex Optimization, chapter 4.7, pp. 174–187. Cambridge University Press, 2004.

---

> > ### Comment · Reviewer_xqpc · 2023-11-21
> > **Re: Official Comment by Authors**
> >
> > Thank you for providing the additional discussions as well as clarifications.
> > They have been helpful and have addressed a number of concerns raised in my original review comments.
> > I suggest the authors provide additional explanations (maybe in the appendix - including useful references such as the blog article the authors referred to in their response) regarding why linear scalarization frequently results in a biased solution and/or fails to find the correct solution based on given scalarization weights, since it may be useful for potential readers.

---

> ### Author Response · Authors · 2023-11-20
>
> Dear Reviewer xqpc,
>
> We express our thanks once more for your valuable time and constructive suggestions in the review process of our work. We have responded to each of your queries in detail. In summary,
>
> - We provide a detailed explanation of the LS optimization scheme from both theoretical and intuitive perspective;
> - We explained why PCI outperforms SOMO+LS with an example when a specific preference, a "small" number of preferences and a variety of preferences are given;
> - We clarified the details of our baseline method I-LS and updated our manuscript;
> - We also clarified that PCI significantly outperforms I-LS，I-LS even harm the surrogate part (DST).
>
>
> If there are any further questions, we are more than willing to provide further explanation. We sincerely hope that our previous responses will be considered in your final evaluation of our work!

---

### Official Review · Reviewer_xC4n · 2023-11-05

**Soundness:** 3 good
**Presentation:** 2 fair
**Contribution:** 3 good
**Rating:** 6
**Confidence:** 2

**Summary:**

This paper proposes a Preference-Conditioned Inversion (PCI) framework to generate Pareto optimal molecules under multiple property requirements. In addition, it provides some theoretical guarantees on the ability of the proposed PCI framework to find the preference-conditioned Pareto optimal solutions. Experiments on benchmark datasets demonstrate the effectiveness of the proposed method.

**Strengths:**

1. This paper proposes a method for multi-objective optimization in discrete chemical space. The proposed method is effective and significantly outperforms baseline methods.

**Weaknesses:**

### Major

1. Some part of the algorithm is not described clearly. See **Questions**.

2. The theoretical guarantee seems weak. It only shows after $T$ optimization rounds, the maximum distance between the molecule obtained and the optimal molecule is bounded.


### Minor

- Section 4.2
"develop an differentiable" -> "develop a differentiable"
- Below Eq(7) "adpot" -> "adopt"

- Section 4.3 "introduce a approach" -> "introduce an approach"

- Algorithm 1, line 11, ".;" -> "."

- Section 5.3 "It demonstrate that PCI..." -> "It demonstrates that PCI..."

**Questions:**

1. In Algorithm 1, how to assemble the scaffolding tree into a molecule? Can you provide some description or references?

2. What is the complexity of solving the linear programming problem Eq. (9)? And what is the total complexity of the algorithm? How does it compare with the complexity of the baseline algorithms?

---

> ### Author Response · Authors · 2023-11-14
>
> Thank you for taking the time to review our paper and providing valuable feedback. We would like to answer your questions below.
>
> **1.  [W1 & Q1] How to assemble the scaffolding tree into a molecule?**
>
>  We follow JTVAE[1], a classic work that transforms molecular graphs into substructure-based tree structures, which has been widely used in drug design. The process is as follows: (a) Ring-atom connection. When connecting atom and ring in a molecule, an atom can be connected to any possible atoms in the ring. Ring-ring connection. (b) When connecting ring and ring, there are two general ways, (1) one is to use a bond (single, double, or triple) to connect the atoms in the two rings. (2) another is two rings share two atoms and one bond.
>
> **2. [Q2] Time Complexity.**
>
> We have provided an analysis of the time complexity in Section B of  appendix. The computation of $C = G^T G$ has runtime $O(m^2 n)$, where $n$ is the dimension  of the gradients. With the current best LP solver [2], our LP (9), that has $m$ variables and at most $2m+1$ constraints, has a runtime of $O^*(m^{2.38})$. Since in deep networks, usually $n \gg  m$, PCI does not significantly increase the computational cost of backpropagation gradient calculation. IIn molecular optimization, the time spent on oracle calls typically dominates the algorithm's complexity.  PCI has the comparable oracle calls to that of DST, it requires $O(TM)$ oracle calls, where $T$ is the number of iterations (Alg 1). $M$ is the number of generated molecules, we have $M \leq KJ$, $K$ is the number of nodes in the scaffolding tree, for small molecule, K is very small. $J$ is the number of enumerated candidates in each node. Furthermore, the change in wall clock time when Pareto optimization is incorporated to baseline method to a initial molecule: $28s \to 63s$.
>
> **3. [W2] About convergence analysis.**
>
> There may be some misunderstandings here. When we talk about convergence, the goal is to discuss the number of iterations required to ensure that the distance between the current point $x_{T}$ and the minimum value point $x^*$ is guaranteed to be smaller enough. In our work, we provide this guarantee as you state ‘
> After T optimization rounds, the maximum distance between the molecule obtained and the optimal molecule is bounded’. This definition of convergence is widely used, including many works about optimization in machine learning. For more detailed information, please refer to https://en.wikipedia.org/wiki/Rate_of_convergence.
>
> **Reference:**
>
> [1] Wengong Jin, Regina Barzilay, and Tommi Jaakkola. Junction tree variational autoencoder for molecular graph generation. In International conference on machine learning, pp. 2323–2332. PMLR, 2018.
>
> [2] Cohen, M. B., Lee, Y. T., and Song, Z. Solving linear programs in the current matrix multiplication time. In Proceedings of the 51st annual ACM SIGACT symposium on theory of computing (STOC), pp. 938–942, 2019.

---

> ### Author Response · Authors · 2023-11-20
>
> Dear Reviewer xC4n,
>
> We again thank you for your valuable time and constructive comments in reviewing our work. We have responded to each of your concerns. In summary,
> - We provided details about how to assemble the scaffolding tree into a molecule;
> - We conducted time complexity analysis, where demonstrate the PCI is comparable to baseline methods;
> - We further explained the misunderstanding about our convergence guarantee;
> - We also revised and updated our manuscript following your suggestions.
>
> Please let us know if you have any further questions. We’d sincerely appreciate it if you could take our previous responses into consideration when making the final evaluation of our work!

---

### Meta-Review · Area_Chair_RGLT · 2024-01-12

**Metareview:**

While I think the authors do a thorough job of responding to many of the reviewers' concerns, the reviewers' comments about a lack of empirical comparison to linear scalarizations I think highlights a general lack of comparison to many existing multi objective black-box optimization techniques. While the authors demonstrate that one particular scalarization fails to find the correct solution in a constructed example, clearly optimizing even a handful of *random* scalarizations would solve the problem in this case, and work has been done on learned scalarizations (e.g. Lin et al., 2022). Furthermore, while Table 2 is extensive, it's not clear whether any of these methods have been adapted in any way to the multiobjective setting  -- e.g. whether GA or BO methods were adapted to use off the shelf, existing multi-objective extensions from the literature.

Ultimately, while the paper is at the borderline, these questions about the strength and appropriateness of the baselines makes it challenging to override the reviewers here.

**Justification For Why Not Higher Score:**

The authors don't really do a great job of comparing to existing multi-objective optimization strategies, and it's not clear which and how baselines (many of which in original papers are single task) were adapted to this setting.

**Justification For Why Not Lower Score:**

N/A

---

### Decision · Program_Chairs · 2024-01-16

Reject